# Geospatial input data for the PALM model system 6.0: model requirements, data sources, and processing

Wieke Heldens[1], Cornelia Burmeister[3], Farah Kanani-Sühring[2,4], Björn Maronga[2,5], Dirk Pavlik[3], Matthias Sühring[2], Julian Zeidler[1], and Thomas Esch[1]

[1]Deutsches Zentrum für Luft- und Raumfahrt, Earth Observation Center, Land Surface Dynamics, Oberpfaffenhofen, Germany
[2]Institute of Meteorology and Climatology, Leibniz University Hannover, Hannover, Germany
[3]GEO-NET Environmental Services GmbH, Hannover, Dresden, Germany
[4]Harz Energie GmbH & Co. KG, Goslar, Germany
[5]Geophysical Institute, University of Bergen, Bergen, Norway

**Correspondence:** Wieke Heldens (wieke.heldens@dlr.de)

**Abstract.** The PALM model system 6.0 is designed to simulate micro- and mesoscale flow dynamics in realistic urban environments. The simulation results can be very valuable for various urban applications, for example to develop and improve mitigation strategies related to heat stress or air pollution. For the accurate modelling of urban environments, realistic boundary conditions need to be considered for the atmosphere, the local environment, and the soil. The local environment with its geospatial components is described in the static driver of the model and follows a standardized format. The main input parameters describe surface type, buildings and vegetation. Depending on the desired simulation scenario and the available data, the local environment can be described at different levels of detail. To compile a complete static driver describing a whole city, various data sources are used, including remote sensing, municipal data collections and open data such as OpenStreetMap. This manuscript shows how input data sets for three German cities were derived. Based on these data sets, the static driver for PALM can be generated. As the collection and preparation of input data sets is tedious, prospective research aims at the development of a semi-automated processing chain to support users in formatting their geospatial data.

*Copyright statement.* TEXT

## 1 Introduction

Nowadays, computational fluid dynamic models are increasingly used to simulate the atmospheric flow within urban environments, e.g., to develop and improve mitigation strategies for heat stress (e.g. Sharma et al., 2018) or air pollution scenarios (e.g. Kurppa et al., 2018). In order to draw a realistic picture of the thermodynamic and dynamic conditions within urban environments, it is required to consider and sufficiently represent all the relevant physics on the micro- and meso-scale, as well as realistic boundary conditions to reflect the real-world conditions. Besides realistic initial and boundary conditions for the atmosphere and the soil, it is crucial to have detailed information on the local environment, i.e., terrain height, building and

street canyon geometries and surface properties, as well as the type and the current state of the plant canopy, in order to accurately represent the real-world conditions within the urban canopy layer. For instance, detailed information on the geometry of the surrounding buildings and the nearby street canyons is necessary to study ventilation and air pollution in street canyons (Lo and Ngan, 2017) or within courtyard cavities (Gronemeier and Sühring, 2019), or to assess the nighttime fresh-air supply by cold-air drainage flows within residential areas.

The PALM model system 6.0, a large-eddy simulation-based code including several components for urban micro-scale simulation, allows the simulation of urban micro climate of realistic urban environments, as it is capable of handling detailed information on the real-world environment (Maronga et al., 2020). PALM is discretized in space using finite-differences, while terrain or buildings are considered via a Cartesian topography. PALM offers several embedded models to simulate physical processes within the urban environment. Namely, this embraces a land-surface model (Gehrke et al., 2020) to consider the surface-atmosphere exchange of heat and moisture at low vegetation-covered, pavement, as well as water surfaces; a building surface model (Resler et al., 2017) to consider the surface-atmosphere exchange of heat and moisture at building walls, windows and intensive as well as extensive green building surfaces; a plant-canopy model to include explicit momentum drag at grid-resolved vegetation as well as evapotranspiration and heating within the canopy; a radiative-transfer model (Krč et al., 2020) to include complex three-dimensional mutual radiative interactions between surfaces and plants; an indoor and building-energy demand model which simulates the amount of released anthropogenic heat by air-conditioning or waste heat; an aerosol model (Kurppa et al., 2019) as well as an air-chemistry model; a biometerorology model (Fröhlich and Matzarakis, 2020) to estimate thermal comfort and UV-exposure; and last but not least a Lagrangian-based multi-agent model to emulate human's behavior and pathways within the complex urban-environment while sampling several air-quality and biometeorological measures.

Using such a complex model, however, entails that the amount and requirements of input data drastically increases compared to simplified scenario studies. For example, to deduce mitigation strategies for heat-wave scenarios for realistic urban environments, the turbulent flow as well as the surface energy balance on the microscale need to be simulated sufficiently accurate. However, radiative transfer processes as well as the partitioning of the available heat into ground as well as surface sensible and latent heat fluxes depend strongly on the type of the local surfaces. Therefore, spatial information about the location of pavements, water bodies, trees, building heights and geometries, etc. is crucial for the simulation of 'real-world' scenarios. The main objective of this paper is therefore to describe the extensive model requirements, data sources and processing of the geospatial input data for PALM 6.0. This paper focuses on the geospatial input data only, which is required to create suitable input files for PALM and that describe the (static) information, representing the local environment and surface boundary conditions for the model. This is a key aspect in the numerical simulation of urban flows since they determine the atmosphere-surface exchange of momentum, heat and moisture.

The paper starts with a description of the input data requirements of PALM (Sect. 2). Data availability, including possibilities and limitations of a wide range of suitable data sources to satisfy these needs are described in Sect. 3. Each of these data sources requires its individual pre-processing to make the data fit to the model input requirements. Sect. 4 demonstrates exemplary how this pre-processing can be realized for three German cities. In Sect. 5 it is described how the input data is prepared for PALM. Sect. 6 provides an example application of PALM using the input data and static driver described in the paper. The paper

concludes with a discussion of the input data and existing challenges that are related to collecting and preparing the input data according to the PALM input data standard.

## 2 Input data requirements by PALM

The geospatial input data for PALM is organised hierarchically, with a set of minimum requirements and further optional input
data, depending on the objective of the simulation and available input data. This section gives a description of the input data requirements of PALM in the different situations as well as a short description of the required parameters.

### 2.1 Requirements and hierarchy

All geospatial input data for the model is provided by the user in a netCDF driver file (hereafter referred to as static driver) that comprises all static (i.e., time-invariant) spatial information as well as metadata according to the so-called PALM input data
standard (PIDS, see Appendix A). The PIDS inherits most of the netCDF Climate and Forecast Metadata Conventions Version 1.7 (CF-1.7)[1] and therefore also conforms to the conventions of the Cooperative Ocean/Atmosphere Research Data Service (COARDS)[2].Depending on the setup (e.g., only dynamic flow or fully thermodynamic simulation with interactive surfaces) there is a minimum set of mandatory variables and several optional ones that need to be included wgbi182wgbi182 in the static driver.

The initialization in PALM follows a multi-step approach, depending on the given level of detail (LOD) of each variable as provided in the static input file. In absence of a static driver, i.e., the lowest level of detail, LOD0, a horizontally homogeneous surface is initialized based on settings using Fortran NAMELIST parameters, e.g., homogeneously vegetated surfaces and surface properties in the land-surface model (see Maronga et al., 2020). In LOD1, surface information is passed to PALM via two-dimensional fields in the static driver. Table 1 gives an overview of all LOD1 fields that can be read by PALM. For
simulations without thermodynamics, i.e., when no interactive surface schemes are used, only the fields `zt` (terrain height) and `buildings_2d` (building height) are used for initialization and at least one of these fields must be provided. Additionally, the field for the building identifier (ID) `building_id` must be set when `zt` is used in order to guarantee a correct mapping of buildings on the terrain (see Sect. 5.2 for details). As the static driver contains rastered data, information about objects that extend over several grid volumes is lost. By using an ID field this information can be retained. Note that `building_id` is
thus also needed when the building-based indoor model of PALM is switched on (see Maronga et al., 2020). For cases with interactive surfaces, each surface element is classified according to its treatment, i.e., default- (i.e., non-interactive), land surface- or urban-type (i.e., building). In setups without interactive surfaces, all surface elements are classified as default-type. In setups with interactive surfaces, a surface classification using the fields `vegetation_type`, `water_type`, `pavement_type`, and `building_type` is utilised (see Sect. 2.6). Currently, each surface pixel $(y, x)$ must be assigned to one of the aforemen-
tioned types. In the future, PALM will also allow a tile approach so that multiple types can be present in one grid box, which

---

[1]cfconventions.org/Data/cf-conventions/cf-conventions-1.7/cf-conventions.html
[2]see ferret.pmel.noaa.gov/Ferret/documentation/coards-netcdf-conventions

**Table 1.** List of LOD1 variables that can be specified in the static driver file.

| Variable name | Dimensions | NetCDF data type | Values/Units | Description |
|---|---|---|---|---|
| albedo_type | $(y, x)$ | NC_BYTE | $0 - 18$ | Optional classification of albedo. Default values are set by building_type, pavement_type, pavement_type, vegetation_type, and water_type but will be overwritten in case albedo_type is defined |
| buildings_2d | $(y, x)$ | NC_FLOAT | m | Building height relative to underlying terrain. Requires setting of building id |
| buildings_id | $(y, x)$ | NC_INT | - | Building id, used to identify single building envelopes for mapping of buildings on complex terrain |
| building_type | $(y, x)$ | NC_BYTE | $0 - 6$ | Bulk classification of building types |
| pavement_type | $(y, x)$ | NC_BYTE | $0 - 16$ | Bulk classification of pavements on soil. Requires setting of a soil type |
| surface_fraction | $(n, y, x)$ | NC_FLOAT | $0 - 1$ | Relative fraction of the respective surface type given via vegetation_type $(n = 0)$, pavement_type $(n = 1)$ and water_type $(n = 2)$. The sum over all relative fractions must be equal to one for each location. Note that more than one surface type per pixel are currently not supported by PALM |
| vegetation_type | $(y, x)$ | NC_BYTE | $0 - 18$ | Bulk classification of non-resolved vegetation surfaces at natural land surface types. Requires setting of a soil type |
| water_type | $(y, x)$ | NC_BYTE | $0 - 18$ | Bulk classification of water bodies |
| zt | $(y, x)$ | NC_FLOAT | m | Terrain height above mean sea level |

will be particularly useful when using coarser grid spacings ($> 10$ m), where neglecting sub-pixel heterogeneity is no longer adequate. The tile approach will be realized by specifying the individual portions via the field surface_fraction, which is already recognized by PALM.

By setting the surface types, all required parameters for the surface treatment are automatically set to default values. Note that pavement- and land vegetation-type surface require the setting of soil_type at the respective pixels. When using the surface classification, a default albedo type is automatically set for each pixel depending on the chosen surface classification.

**Table 2.** List of LOD2 variables that can be specified in the static driver file.

| Variable name | Dimensions | NetCDF data type | $n$ | Description |
|---|---|---|---|---|
| albedo_pars | $(n, y, x)$ | NC_FLOAT | $0 - 7$ | Optional classification of individual albedo values for broadband, longwave, and shortwave radiation for each pixel $(y, x)$. |
| buildings_3d | $(z, y, x)$ | NC_BYTE | 0-1 | Three-dimensional building topology relative to underlying terrain in which setting of 1 refers to building and 0 refers to no building. The $z-$ dimension only needs to be large enough to embrace the building topology |
| building_pars | $(n, zwall, y, x)$ | NC_FLOAT | $0 - 46$ | Optional setting of building material parameters for each wall layer $zwall$ and pixel $(y, x)$. |
| building_surface_pars | $(n, y, x)$ | NC_FLOAT | $0 - 46$ | Optional setting of building surface parameters for each pixel $(y, x)$. |
| pavement_pars | $(n, zsoil, y, x)$ | NC_FLOAT | $n = 0 - 1$ | Optional setting of pavement thermal parameters for each soil layer $zsoil$ and pixel $(y, x)$. |
| pavement_surface_pars | $(n, y, x)$ | NC_FLOAT | $n = 0 - 3$ | Optional setting of pavement surface parameters for each pixel $(y, x)$. |
| root_fraction | $(zsoil.y, x)$ | NC_FLOAT | $n = 0 - 1$ | Root fraction within the individual soil layers $zsoil$ |
| soil_pars | $(n, zsoil, y, x)$ | NC_FLOAT | $n = 0 - 8$ | Optional setting of soil parameters for each soil layer $zsoil$ and pixel $(y, x)$. |
| vegetation_pars | $(n, y, x)$ | NC_FLOAT | $n = 0 - 11$ | Optional setting of vegetation parameters for each pixel $(y, x)$. |
| water_pars | $(n, y, x)$ | NC_FLOAT | $n = 0 - 6$ | Optional setting of water body parameters for each pixel $(y, x)$. |

This can, however, be overwritten using the optional field albedo_type. Tables A1-A7 in the Appendix give an overview of the classifications used and the parameters automatically set when using LOD1.

Based on the LOD1 classification of each surface pixel, the static driver allows to overwrite all or selected parameters that were automatically set by the LOD1 input data (e.g., leaf area index (LAI), surface emissivity, etc., see Tables A1-A7 in the Appendix). For each *_type field in LOD1 there is thus a respective *_pars field, representing LOD2 data (see Table 2). Note that LOD2 can only be used when simultaneously having specified LOD1 data. The *_pars fields then can contain fill values except for those locations where the data should be overwritten by LOD2 input data. Figure 1 shows

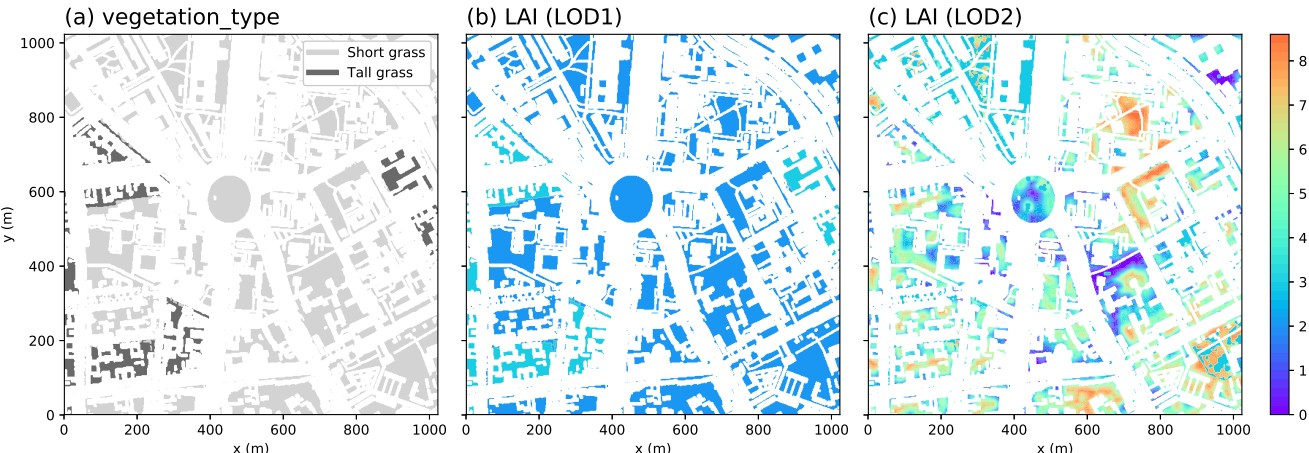

**Figure 1.** Illustration of LOD1 and LOD2 hierarchy for the LAI around Berlin Ernst-Reuter-Platz. Panel (a) shows the determined `vegetation_type` classification, which involves automatic setting of an LAI for all pixels classified as vegetation (LOD1), shown in panel (b). Panel (c) shows LOD2 information of the LAI distribution transferred by the field `vegetation_pars`.

**Table 3.** List of LOD3 variables that can be specified in the static driver file.

| Variable name | Dimensions | NetCDF data type | $n$ | Description |
|---|---|---|---|---|
| building_surface_pars | $(n, z, y, x)$ | NC_FLOAT | $0-46$ | Optional setting of building surface parameters for each surface height $z$ and each surface pixel location $(y, x)$. |

this hierarchy exemplary for the LAI based on available data for Berlin, Germany. Additionally, LOD2 offers the NC_BYTE field `buildings_3d`, which can be used to specify three-dimensional building structures including overhanging structures, thoroughfares, and bridges (see Sect. 2.5). Unlike for the other `*_pars` fields, the LOD1 data (i.e., `buildings_2d`) is not used if LOD2 data (i.e., `buildings_3d`) is present in the data. Furthermore, the field `root_fraction` can be set in order to specify a different vertical root distribution in the soil model of the parameterized vegetation in the land surface model.

**Table 4.** List of LOD4 variables that can be specified in the static driver file.

| Variable name | Dimensions | NetCDF data type | $n$ | Description |
|---|---|---|---|---|
| `building_pars` | $(n, zwall, s)$ | NC_FLOAT | $0 - 46$ | Optional setting of building material parameters for each wall layer $zwall$ and each surface element $s$. |
| `building_surface_pars` | $(n, s)$ | NC_FLOAT | $0 - 46$ | Optional setting of building surface parameters for each surface element $s$. |

While LOD2 is limited to a localized setting of individual surface or material properties based on location $(y, x)$ only, LOD3 and LOD4 settings (see Tables. 3 and 4) allow an even more detailed specification of building parameters. Note that in LOD4, the input data no longer depends on the rastered PALM grid, but is arranged in a one-dimensional array of size $ns$, where $ns$ is the number of surface elements on the model domain. For each surface element, the user then has to specify the position of the surface element in the PALM domain space, i.e., $(z(1 : ns), y(1 : ns), x(1; ns))$ as well as the orientation of surface elements in terms of azimuth and zenith angles ($azimuth(1 : ns)$ and $zenith(1; ns)$, respectively) in one-dimensional fields.

Additionally, three-dimensional fields of leaf-area density, LAD (`lad`), and basal-area density, BAD (`bad`, implementation under development), as well as vertically distributed root fractions (`root_fraction_resolved`), and a tree ID (`tree_id`) can be used to set-up resolved-scale plant canopies (see Sect. 5.3).

## 2.2 Geo-referencing

Various model components such as the radiation parametrization, the representation of the Coriolis force or geo-referencing of model output require information about the geo-location of the grid cells of PALM. Therefore, the static input file must contain information about the longitude and latitude, as well as the Easting and Northing UTM coordinates of the lower-left corner of the model domain. Furthermore, reference height of the lowest model grid point as well as the rotation angle of the model domain must be provided, which is especially important to setup virtual measurement positions and trajectories within the model according to 'real-world' measurements (Maronga et al., 2020). The required coordinate information must be given as global attributes in the NetCDF file.

## 2.3 Terrain height

To consider effects of elevation changes on the flow, the terrain height `zt` can be provided for each discrete $(y, x)$-location in the model. Data gaps leading to fill values are forbidden. In case `zt` is not provided, the land surface is set-up at $zt = 0$ m.

The `zt` can be provided in absolute values, i.e., in meters above sea level, or in relative heights where, e.g., its minimum value is already subtracted. If absolute values are used, PALM will subtract the minimum value within the domain itself to save computational grid points (no computations are needed within the soil). At this point we note that the original terrain height

might be further processed and slightly modified by PALM to fulfill certain requirements, which is described in detail in Sect. 5.2.

## 2.4 Surface classification

In order to parameterize atmosphere-surface-interactions, PALM needs to solve the energy balance at physical surfaces. To
do this, several physical surface parameters such as heat capacity, roughness, albedo, emissivity, information about vegetation, etc., must be known. To allow for proper LOD1 initialization of material parameters and surface properties via pre-defined lists, PALM classifies all horizontal and vertical surfaces in the model according to their general type, e.g., whether it is a building or a vegetation surface. PALM considers four different types of surfaces: building, vegetation, pavement and water surfaces, while the surfaces are classified in a two-step approach. In a first step, grid points are flagged as atmosphere, building or terrain
grid point. Surfaces which belong to a building grid point are automatically flagged as building surfaces, while surfaces which belong to a terrain grid point are flagged as land surfaces. In a second step, surfaces are further specified according to their respective type, which enables proper LOD1 initialization with pre-defined lists for material and surface properties. For this reason, input of `building_type`, `vegetation_type`, `pavement_type` and `water_type` is required. At each $(y, x)$-location at least one of these types must have a non-missing value, so that each surface element can be classified appropriately,
either as a pavement, vegetation, or water. It is also required that the given `*_type` matches with the general classification into building- and land-surface, i.e., at locations where buildings (see Sect. 2.5) are defined also `building_type` must be defined, while at land-surfaces at least one one of `vegetation_type`, `pavement_type` or `water_type` must be defined.

## 2.5 Buildings

Information on the location and height of buildings can be provided as two-dimensional buildings heights (`buildings_2d`, LOD1) or as a three-dimensional integer array (`buildings_3d`, LOD2), where each building (non-building) grid point is masked by 1 (0). At locations where no buildings are located, `buildings_2d` may contain fill values, while `buildings_3d` must not contain any fill values. In the LOD1 case, buildings are always mounted on the Earth's surface and overhanging structures such as tunnels or bridges are not allowed, while in the LOD2 case also overhanging obstacles are allowed. In both cases,
building information is given relative to the terrain height and buildings are mapped onto the top of the terrain during model initialization, which is described in detail in Sect. 5.2. At this point we note that PALM can also consider bridges, which can be input as three-dimensional building structures but require a special treatment as further discussed in Sect. 5.2.

To distinguish between single buildings, e.g., in order to map them accordingly onto the underlying terrain (please see Sect. 5.2), or to compute the energy demand of single buildings (Maronga et al., 2020), each building has an unique identifica-
tion number (`building_id`) that must be given in the static input file at each $(y, x)$-location where `buildings_2d` or `buildings_3d` is defined.

To solve the energy balance at building surfaces (Resler et al., 2017; Maronga et al., 2020), information on the type of the buildings must be provided. This includes information on various wall material and surface properties must be known, e.g.,

wall thicknesses, heat capacities and conductivities, window and wall fractions, albedo, etc. which depend on the individual construction parameters and the current state of building restoration. As many of these information are quite often unknown, buildings are classified into characteristic types in order to use default parameters. For this, buildings are classified according to their year of construction and its general usage (residential or office building). PALM provides lists of wall material and surface parameters for six building types (see also Tab. A2): 1 - residential buildings built before 1950, 2 - residential buildings built between 1951 and 2000, 3 - residential buildings built after 2001, 4 - office buildings built before 1950, 5 - office buildings built between 1951 and 2000, and 6 - office buildings built after 2001, and 7 - bridges. At this point we note that `building_type` = 7 is exclusively used to identify bridges and to distinguish them from other three-dimension building structures. The respective `building_type` must be provided for each discrete $(y, x)$-location where `building_2d` or `building_3d` is defined. The provided wall and surface parameters are assumed to be valid for Germany, but still have to be checked and likely changed for other building styles. Therefore, even though building parameters are difficult to aggregate in practice, PALM allows to prescribe different types defined among a single building, e.g., to consider building extensions with different physical properties or different usage in case such information is available. In addition, to modify wall material and surface parameters at different (y,x)-location or even at single surface elements, `building_pars` or `building_surface_pars`, respectively, can be optionally provided.

## 2.6 Land surfaces

Beside the general classification via pre-defined parameter lists for each land-surface type, physical surface parameters can be further specified with `vegetation_pars`, `water_pars` or `pavement_pars`, which can be optionally provided, as explained in Sect. 2.1.

### 2.6.1 Vegetation

PALM distinguishes between parameterized vegetation that is not resolved by the numerical grid (vegetation height smaller than the vertical grid spacing) and thus considered flat (e.g., short grass), and tall vegetation that can be partially resolved by the numerical grid, depending on the grid spacing used (e.g., shrubs or trees). Parameterized vegetation is considered within the energy balance solver for land surfaces, where the given `vegetation_type` defines the physical properties at the respective surface element, e.g. LAI, typical vegetation coverage of the soil, roughness, heat conductivity between the soil and the skin layer, emissivity, etc.. Currently 19 vegetation classes are defined in PALM, such as crops, shrubs, forests, etc. (see Tab.A1). These can optionally be specified in more detail by `vegetation_pars`, which may contain missing values for single parameters and locations, and the respective properties are only updated and customized where they contain non-missing values, i.e., it is allowed to provide parameters only at locations where these are available. At parameterized vegetation surfaces, additional information concerning the root-area-density distribution (`root_area_dens_s`) within the soil can be optionally provided. If it is not provided it is taken from bulk parameter lists defined by the given `vegetation_type`.

In contrast to parameterized vegetation, resolved vegetation directly accounts for a sink term in the momentum equations (e.g. Kanani-Sühring and Raasch, 2015) and directly affects its surroundings via shading and three-dimensional reflections

(Resler et al., 2017). To consider these effects in the model, information about the leaf-area density (LAD) within the respective grid volumes is required and can be input via `LAD`, which is mapped on top of the underlying terrain. The leaf-area density in the model is initialized at every location where `LAD` has non-missing and positive values, elsewhere it is set to zero.

### 2.6.2 Pavements

Pavement surfaces can be specified via `pavement_type` which defines the surface and subsurface material properties via pre-defined parameter lists, which currently include 15 pavement types such as asphalt, concrete or cobblestone (see also Tab. A3). Pavement parameters are for example heat conductivities and capacities at different pavement layers, surface roughness lengths and emissivity, etc.. To further specify pavement parameters, `pavement_pars` and `pavement_subsurface_pars` can be optionally provided for defining surface and sub-surface parameters, respectively.

Also, it is possible to provide additional input for street types and street crossings with `street_type` and `street_crossing`, respectively. The street-type classification (e.g. highway, local road, pedestrian road etc.) can be used by PALM to parameter-ize traffic emissions within the embedded chemistry model, while both the street network and crossings may in the future be employed by the embedded multi-agent system for urban residents (Maronga et al., 2020). The crossings parameter therefore indicates the area where an agent is allowed to cross the streets.

### 2.6.3 Water bodies

Water surfaces can be specified via `water_type` which defines the surface properties via pre-defined parameter lists (see Table A5 in the appendix). Water types included are ocean, lake, river, pond and fountain. To further specify water surface parameters, `water_pars` can be optionally provided. Here the water temperature can specified, as well as $z_0$ and $z_0, h$. Emissivity of all water types is 0.99 and they belong to `albedo_type` 1, which is also specified in `water_pars`.

### 2.7 Soil classification

To consider the interaction of the land surface with the underlying soil at vegetation and pavement surfaces, a `soil_type` must be given at grid cells that are classified as vegetation or pavement surfaces, defining a list of default physical parameters for pre-defined soil types, based on the granularity of the soil (e.g. coarse, medium, fine, or organic soil). `soil_type` can be given for different level of detail. For LOD1, `soil_type` must be provided for each $(y, x)$-location where vegetation or pavement is

defined, assuming that soil properties are vertically homogeneous, while for LOD2 `soil_type` is given for each $(z_{soil}, y, x)$-location, with $z_{soil}$ being the depths of the soil layers, in order to consider variations of soil properties also in the vertical direction. Respective soil properties are e.g. the Van Genuchten parameter, hydraulic conductivity, volumetric soil moisture at saturation, field capacity at wilting point or the residual soil moisture. By default the soil layers have a depth of (from top to bottom) of 0.01, 0.02, 0.04, 0.06, 0.14, 0.26, 0.54, and 1.86 m, where pavement layers are by default assumed to cover the six

uppermost layers (see also Tab. A4). Note that the vertical composition of materials used in road construction varies a lot and is usually unknown. The current implementation thus should be considered to be a first guess. To further customize physical

soil parameters, `soil_pars` can be optionally provided either as LOD3 to provide vertically homogeneous parameters, or as LOD4 to provide vertically heterogeneous parameters, i.e., each soil layer at each relevant surface element can be given individual physical properties. `soil_pars` may contain missing values and is only used to update the physical soil parameters at locations and for parameters that are non-missing, i.e., it is allowed to provide only single parameters at locations where this
information is available.

## 2.8 Surface albedo

Information concerning the albedo for the different surfaces is already provided within the pre-defined parameter lists for building and land-surfaces. However, more detailed information concerning the `albedo_type` (predefined list of broadband and spectral albedos for direct and diffuse radiation) or broadband / spectral surface albedos at each $(y, x)$-location
(`albedo_pars`) can optionally be provided.

## 3  Data sources

Masson et al. (2020) review various data sources for urban climate models at meso- and micro-scale. The requirement of a spatial resolution of 1 - 10 m for building-resolving simulations, being a key focus of PALM, reduces the available sources of data significantly. The most important sources are remote sensing, governmental/municipal data and open data, as they allow
area-wide and automated pre-processing, while field surveys and manual mapping are only a practicable option for small areas of interest, e.g. of one building block in a city. Often, a combination of different sources is required to achieve a consistent coverage with detailed information for the entire area of interest.

The observation of the characteristics and dynamics of the Earth's surface by means of remote sensing has become increasingly important in recent years. In general, remote sensing approaches take advantage of the fact that material- or object-specific
interactions occur between the surface and land cover type on the one hand, and the electromagnetic radiation interacting with them on the other hand. This specific spectral signature or back-scattering pattern can then effectively be used to identify and discriminate different surface and material types. Active imaging systems such as radar or laser scanners carry their individual radiation sources (Baghdadi and Zribi, 2016). The intensity and pattern of the backscattering then allows mapping the position, type and, in case of laser scanners, height of surfaces and objects. This can be used to create digital surface models, e.g., based
on the radar satellite TerraSAR-X at a global scale (Rizzoli et al., 2017) or at local scale using airborne LiDAR systems (Yan et al., 2015). Optical remote sensing makes use of the reflected radiation of the sun. There is a broad range of systems available, mounted on satellite platforms as well as airborne and UAV-mounted sensors. The selection of the sensor used depends on spectral characteristics, spatial resolution, availability for the area of interest and costs. Typical mapping tasks carried out with optical remote sensing are land cover mapping (e.g. Khatami et al., 2016; Wulder et al., 2018) and vegetation characterization
(e.g. Verrelst et al., 2015). As the PALM model requires high spatial resolution for performing building-resolving simulations, the free and open Sentinel-2 satellite data is of interest as well as the data of commercial satellite constellations like Rapid Eye

or World View. Additionally, false color airborne imagery, with its very high spatial resolution,would be preferable if available for the right time of year.

Especially in developed countries, public authorities and agencies routinely collect a vast amount of geo-spatial data sets. The following focuses on the situation in Germany, because the selected study areas for the model development are located here. In Germany, available official data is hosted at different levels of agencies and departments (municipal, federal state, state, cadastral office, etc.). The accessibility of the data differs between the federal states and municipalities. In some federal states, such as Berlin, Hamburg, Thuringa or North-Rhine-Westfalia the data is easily accessible and downloadable and available through an Open Data Licence. These data sets are also regularly updated. The possibility to use additional official data depends on the purpose and costs. Municipalities interested in the resulting micro-climate simulations usually provide their data on request for the purpose of a scientific study. The various German official data sets which are useful and available for most municipalities are addressed below. The cadastral data of ATKIS and/or ALKIS (Working Committee of the Surveying Authorities of the States of the Federal Republic of Germany, 2015) are used to estimate the building age when no detailed information for individual buildings are available. Additionally, the data set can be used for the localization of streets, public open spaces and water bodies. ATKIS and ALKIS are regularly updated every one to three years (depending on the land use category). The municipal parks and open spaces departments host the data of the public green spaces and tree register. The latter is usually only available for trees on public land. Information about trees and green spaces on private property have to be derived from additional data sources. If a tree register is available it provides comprehensive information on tree species, age, height, and sometimes also crown and stem diameters. Building data is provided in form of 3D building models in level of detail LOD1 (block buildings without exact roofs) or LOD2 (more detailed with roof parts). Since 2019, LOD2 building data exists for all German states (Arbeitsgemeinschaft der Vermessungsverwaltungen der Länder der Bundesrepublik Deutschland, 2019a, b). However, accessibility and cost vary. A standardized data format for 3D city models is City Geography Markup Language (CityGML, Open Geospatial Consortium (2012)), an XML based data format that can be used to describe the city in 3D at different levels of detail. Digital terrain, surface models or LiDAR data as well as aerial images are available at the departments of geo-information or land survey administration at the federal state level in general. Aerial images are updated in a 2 to 5 years period to monitor the green volume development. For this purpose the images have to include the near infrared band. The acquisition dates differ - dependent on their primary purpose- from early spring to summer and thus have a minimal or dense broad leave cover. Only the summer images present the phenological state needed to detect the tree canopy and with that the LAI and leaf area density (LAD). Soil data is available at the municipal level or at the state level in different scales from 1:10000 to 1:200000. Generally the municipal data cannot provide the full information required for a model parameterization, so that additional data acquisitions and/or data fusion is needed.

Surprisingly, municipalities – at least in Germany – usually don't systematically collect spatially detailed information on the road network and pavement types. This gap can be closed using Volunteered Geographical Information from the Open-StreetMap (OSM) project. One caveat of such crowd driven data collection is that anybody can add any features and tags they think relevant, so no homogeneous data quality, completeness and adherence to a single standard can be guaranteed (Quinn and Bull, 2019). However Haklay (2010) and Graser et al. (2015) show that at least in western Europe OSM has a data quality en

par with governmental sources. OSM can be utilized to add missing information to the government data e.g., about road type, pavement type, bridges, pedestrian crossing points, and water bodies.

Geodata can be stored in raster format, with a value for each raster cell or pixel. GeoTIFF is a common format supported by all geo-processing software, but also the NetCDF supports geo-spatial information. Spatial data in vector format uses the locations of point, lines and polygons with optionally attached attribute tables containing additional information on each spatial object. A commonly used vector format is the ESRI Shapefile. Governmental data are often in vector format, as there are many attributes that describe an object. Remote sensing data are mainly raster data, as such data is recorded by the sensor in a regular grid.

## 4 Pre-processing of input data

Existing approaches to generating surface parameters for urban climate analysis, such as local climate zones (LZC) (Stewart and Oke, 2012) (e.g. as implemented in the WUDAPT project (Ching et al., 2018)) or the MApUCE tool (Bocher et al., 2018) focus on a coarser scale than PALM, as most urban climate studies do as well. The LCZ concept was considered not detailed enough to use it as basis for the generation of input data, as it provides indicator values per neighbourhood of 100 - 500 m, with no or little information on the configuration within the area. WUDAPT also recognises that this is sub-optimal for grid based modelling applications and extends their research to mapping urban morphology parameters at finer resolution (30 - 100 m) using Landsat satellite data and the LCZ concept (Zonato et al., 2020). The simulation results of a numerical weather prediction model (NWP), in this case the WRF (Weather Research and Forecast) model, is according to Zonato et al. (2020) similar to Lidar derived data at the scale of 0.5 km and coarser. This is a very promising result, but as the spatial resolution of PALM is within the range of several meters, more detailed solutions have to be implemented for PALM. The project MApUCE (Bocher et al., 2018) provides a framework for generating urban indicators in a standardized way, in this case for all French municipalities at different levels of scale, from building to district. Although standardized mapping of indicators would be very helpful also for PALM, this system would need to be adapted thoroughly to allow for sub-building indicators that correspond to the grid cells of PALM instead of buildings or building blocks.

This section introduces a strategy and workflow for the selection and pre-processing of the input data sources for PALM 6.0. This is demonstrated for three cities in Germany with varying availability of input data: Berlin, Stuttgart and Hamburg. Despite the variety of data sources, it was aimed to automate the pre-processing for each layer as much as possible to ensure replicability and to handle the vast volume of data (0.5 – 1 TB per city, depending on target resolution and city area). This resulted in a collection of pre-processing scripts to which adaptations have been made for each of the three cities.

The municipal data of Berlin including the aerial imagery and 3D building model was retrieved from the Berlin Geoportal FIS Broker[3]. The municipal data including the aerial imagery and 3D city model Hamburg was retrieved from the Transparenzportal of the governemental offices in Hamburg[4]. The municipal data including the aerial imagery and 3D building model of Stuttgart

---

[3]https://fbinter.stadt-berlin.de/fb/index.jsp

[4]http://transparenz.hamburg.de/das-transparenzportal/

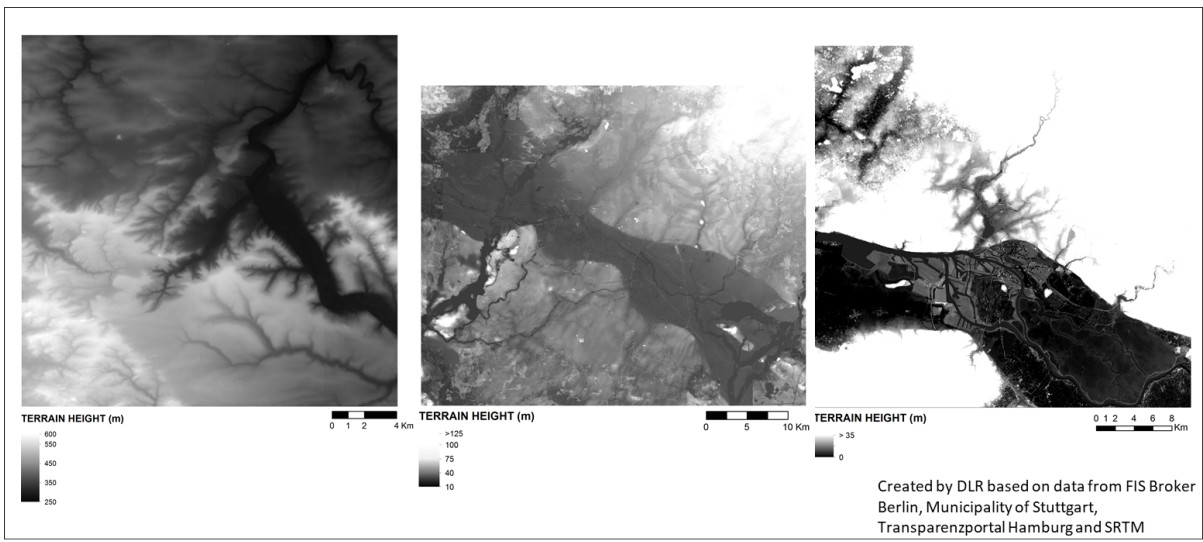

**Figure 2.** Terrain height for the study areas in m above ground. From left to right: Stuttgart, Berlin and Hamburg.

was provided by the Landeshauptstadt Stuttgart for use in the [UC]² project (Scherer et al., 2019). Other sources of data are indicated directly in the text.

The scripts and programs used to process the input data are currently not publicly available as they use some commercial and/or internal functionality. However we are currently in the process of maturing them and plan to make them publicly available with the PALM Model in the future.

## 4.1 Terrain height

Active remote sensing systems are valuable sources to generate digital surface models. For the layers of Berlin, Stuttgart and Hamburg, products of two different active sensors are combined. Within the municipality boundaries, the terrain height is directly retrieved from the digital terrain model (DTM) data derived from airborne LiDAR data as provided by each of the municipalities in 1 m horizontal resolution [5] [6]. As this data set ends exactly at the municipal boundaries, a satellite based, yet coarser data set is added to provide terrain height for the surrounding areas, as PALM always requires a rectangular model region. In this case, the 30 m SRTM digital elevation model (DEM) was used. It is derived from the Shuttle Radar Topography Mission (SRTM) (Farr et al., 2007). For Stuttgart and Hamburg, the SRTM data set first had to be transformed from the global lat/lon geoid to the local German geoid. Subsequently, the SRTM DEM was clipped and resampled to the study areas with 1 m spatial resolution. Finally, the local terrain model and the SRTM terrain model were merged, with the local terrain model as primary source. A feathering distance of 100 pixels was assigned for borders of the local terrain model to smooth any abrupt changes in height between the two data sets. The final terrain height data set for each of the three cities is shown in Fig. 2.

---

[5]Berlin: https://fbinter.stadt-berlin.de/fb/feed/senstadt/a_dgm
[6]Hamburg: http://suche.transparenz.hamburg.de/dataset/digitales-hohenmodell-hamburg-dgm-16

## 4.2 Surface classification

PALM differentiates between building and land surface grid cells, where the land surface grid cells must consist of vegetation, pavements or water bodies (see Sect. 2.4). The first task within the surface classification is to map these four classes (buildings, vegetation, pavements and water bodies). The sections below describe how the according maps are prepared so that they can be written out into the static driver file. As soon as the four class rasters are available, possible gaps, which usually result from combining data sets from different sources or an artefact of rasterization, need to be filled, in at least one of the core classes (see Sect. 2.1). To achieve this, the layers were ranked according to their spatial reliability, e.g, the building layer was preferred over the – often courser – vegetation layer. Secondly, extra secondary buffered input layers were generated where possible and used to fill in their primary layer for pixels were none of the primary layers or prior filled layers had valid data. For example this was necessary for roads, where exact information on roadside parking was not available and thus the actual paved surface is wider than what would be expected from the road width. If there were still holes after all the filling iterations they were filled with a prevalent reasonable value like bare soil, which is internally considered as a `vegetation_type` and parameterized accordingly (see Tab. A1).

After the general surface classification is done, unique IDs for each of the buildings and bridges are generated to mark which pixels belong to the same object to support processing in PALM (see Sect. 5.2).

## 4.3 Buildings

For all the cities the building height, building type and building IDs were derived, which is described in the following sections.

### 4.3.1 Building height

For realistic simulation results of both the flow and the thermodynamic interaction with the urban canopy, it is essential to have the spatially resolved and correct building height. In Germany municipalities have 3D building outlines in LOD1 (block model) or LOD2, which contains differentiated roof structures and therefore allows spatially explicit height calculation for each pixel. For Hamburg and Berlin LOD2 data was available as CityGML (Open Geospatial Consortium, 2012) data[7,8], while in Stuttgart LOD2 building height data was provided as 3D triangulated irregular network (TIN). The developed approach to calculate the building height is a two step approach. In a first step the 2D-coordinates of all pixel centroids inside a single polygon as well as the 3D-bounding box of the polygon is calculated using the algorithm from the GDAL Library GDAL/OGR contributors (2019), which can cope with complex building geometries including inner courtyards etc. If the polygon is of single height, which is (nearly) always the case for floor polygons, this single height is used for all pixels. For each single centroid-coordinate the 3D intersection between its vertical line and the plane of the polygon is calculated to get the height of the building at this position. The special cases that the vertical line is inside or parallel to the plane (building walls directly passing through pixel centroids) are filtered. The calculated height values are capped to the $z$-range of the bounding box. This

---

[7]Berlin: https://fbinter.stadt-berlin.de/fb/feed/senstadt/a_lod2

[8]Hamburg: http://suche.transparenz.hamburg.de/dataset/3d-stadtmodell-lod2-de-hamburg4?forceWeb=true

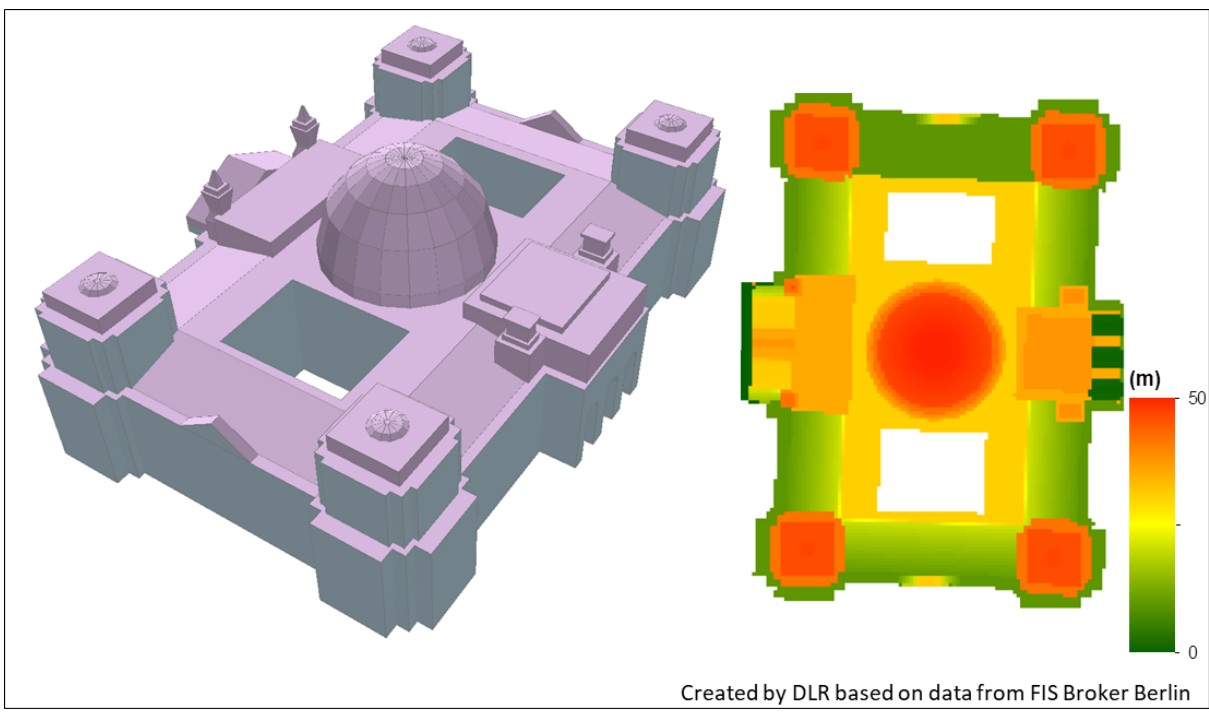

Created by DLR based on data from FIS Broker Berlin

**Figure 3.** 3D visualistion of the CityGML data of the Reichstag building in Berlin (left) and the derived building height (right).

is necessary to compensate for rounding errors in nearly vertical planes, which could lead to single intersects wrongly being near infinite. The minimal and maximal height intersection is stored for each pixel. This approach works on the assumption that all the single polygons are planar polygons as defined in the cityGML standard (Open Geospatial Consortium, 2012), where all points of a polygon are in the same plane. However, for some single buildings in Berlin this assumption proved

wrong as roof planes included single points from the walls of floors. These polygon errors were corrected where possible and otherwise removed from further calculations. Once all polygons have been processed the building height is calculated as the difference between max and min intersection. In a second iteration the same approach is repeated for all pixels that intersect the boundary of the polygons, but where the centroid is outside the polygon. However, these height values are only used for pixels where no building height was calculated in the first iteration. This two step approach guarantees that the extrapolation (with

capping at the z-range of the polygon) to coordinates outside the building footprint is only performed if no other polygons contains the centroid of this pixel. Therefore a higher building which slightly intersect the corner of a pixel will not interfere with lower buildings that cover the pixel, while still removing a lot of single pixel holes with the other base layers (water, vegetation, pavement). This approach worked well for the city of Hamburg and Berlin, where some broken polygons hat to be fixed beforehand. However in Stuttgart not all buildings had a closed floor polygon. Therefore in this case the terrain height

was used as the lower boundary of the buildings. An example of the building height from the CityGML data of Berlin is given in Fig. 3 for the Reichstag building.

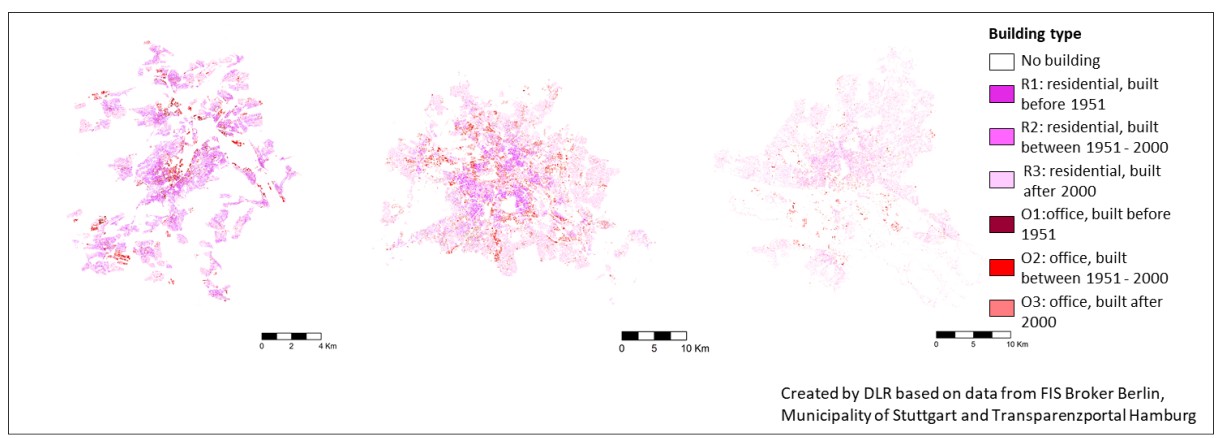

**Figure 4.** Building type maps of Stuttgart, Berlin and Hamburg (from left to right).

### 4.3.2 Building type

The building type describing the energetic properties of the buildings used in PALM is defined through a combination of building use and age (see Tab.A2). In Germany no cadastral information on restoration, fassade changes and heat insulation actions for individual buildings are available, so that the age of building construction used for building classification is often
a rather poor proxy for the thermodynamic properties of buildings. Theoretically, energetic information on buildings exists in Germany, described in the energy performance certificate of each residential building. However, this information is not generally accessible. Currently, a research activity funded by the German Federal Ministry of Economic Affairs and Energy is developing a data base on energetic characteristics of non-residential buildings (ENOB DataNWG[9]). Thus, for small areas where high quality simulations are desired, field surveys or drone surveys are still the best choice. For the large area applications
in this study however, this was not possible. Instead, we aimed at identifying the classes defined in Tab. A2. In Germany, the municipalities often maintain a building use data base, e.g., in the ALKIS data sets. Usually this data is provided at building block level and therefore often contains mixed uses. For Stuttgart and Hamburg this data set was used to distinguish between residential and other building use[10]. The lookup table is given in Table B8. The cities of Hamburg and Stuttgart additionally maintain a data base documenting the age of the building. This allowed the building type to be assigned with quite high
reliability. For Berlin, a combined data set is available, where building blocks are categorised by use and building construction period at the same time[11]. The look-up table to translate this information to building types is listed in Table B7. The building type maps for Stuttgart, Berlin and Hamburg are presented in Fig. 4.

---

[9]https://datanwg.de/home/aktuelles/ (in German)

[10]Hamburg: http://suche.transparenz.hamburg.de/dataset/alkis-ausgewahlte-daten-hamburg6?forceWeb=true

[11]https://fbinter.stadt-berlin.de/fb/wfs/geometry/senstadt/re_isu5

## 4.4 Vegetation

PALM can handle very detailed information on the vegetation, as outlined in Sect. 2.6.1. In this study, the vegetation type was determined as well as vegetation on roofs and several characteristics of trees. As an area wide approach satellite or airborne imagery provides accurate information on the location of the vegetation. Such remote sensing data can also provide estimates for some vegetation characteristics, but not all required by PALM. Luckily, in Germany there is a huge amount of information available from municipal data that cannot be retrieved with remote sensing data alone. Also, open data such as OSM and other citizen science projects can provide valuable information on the urban vegetation. Therefore these sources are all combined to derive as complete input data for PALM as possible.

### 4.4.1 Vegetation type

For the vegetation type layer, municipal data was used in the three demo-cities, including ALKIS (Berlin) and the Biotope Cadastre (Hamburg). For Stuttgart, such municipal data was not available, thus OSM was used as the main source here. Subsequently, gaps were filled with data from Corine Land Cover (CLC, European Union (2017)), which was especially the case for areas outside the municipal borders. Missing data and gaps in the layer between vegetation and other features are filled up using aerial color and infrared images (CIR images), using a threshold on the Normalized Difference Vegetation Index (Rouse et al., 1974) (NDVI) to differentiate between grass and trees. The NDVI can be calulated by Eq.1:

$$NDVI = \frac{\rho_{nir} - \rho_{red}}{\rho_{nir} + \rho_{red}} \tag{1}$$

where $\rho_{nir}$ is the reflection in near infrared part of the spectrum and $\rho_{red}$ the reflection in the red part of the spectrum. Pixels with a NDVI between 0.2 and 0.42 where assigned to vegetation class 3 (short grass), pixels with NDVI values above 0.42 are assigned to class 7 (decidous broadleaf). The thresholds have been selected empirically based on the available air born imagery. Depending on sensor, date, time of day, weather at time of the acquisition or pre-processing options, optimal thresholds might differ for other cities.

For the different data sources lookup tables have been created to map the classes of OSM[12] (Table B2), CLC (Table B2) and the biotope maps of Hamburg[13] (Table B3) to the vegetation types of PALM. The main classes of ALKIS already directly match the PALM vegetation types. The vegetation type layer was created together with other layers as described in Sect. 4.2. As a result, the vegetation type layer is empty in locations of streets and water bodies. The result is shown in Fig. 5.

### 4.4.2 Vegetation on roofs

Intensive and extensive green roofs are detected using municipal 10 to 20 cm Ortho near infrared (CIR) images in combination with the building footprints for the cities of Berlin and Stuttgart. The majority of green roofs are extensive green roofs which mostly have a shallow substrate of 100mm to 250mm and are usually planted with low maintenance water stress tolerant

---

[12]https://planet.openstreetmap.org/
[13]http://suche.transparenz.hamburg.de/dataset/biotopkataster-hamburg1?forceWeb=true

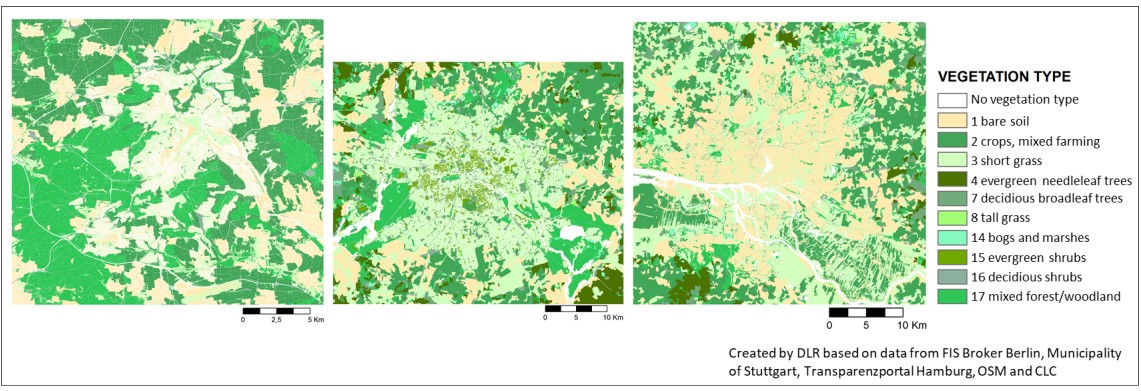

**Figure 5.** Vegetation type. From left to right: Stuttgart, Berlin and Hamburg.

mosses, succulents, herbaceous plants and grasses. Intensive green roofs are rarer and mostly belong in the category of recreational roof top parks. They consist of gardened landscape of grass, shrubs and even small trees (FLL, 2002). The percentage of green roof vegetation is aggregated for each building roof pixel. The mostly very extensive green vegetation on roofs is detected by analysing the NDVI (Eq. 1). The NDVI utilizes the unique characteristic of photosynthetic active vegetation to
absorb light in the red part of the spectrum and emit it in the near infrared, making vegetation distinguishable from other materials. Minimum and maximum thresholds on the NDVI were determined empirically from the aerial images, as the thresholds depend on the vegetation conditions during image acquisition, the pre-processsing and colour enhancing steps during the creation of the images as well as the sensor-systems used aboard the plane. Therefore the thresholds vary between cities. The lower threshold (0.2 in Berlin, 0.06 in Stuttgart) distinguishes the extensive vegetation from bare roofs, while the upper bound
(Berlin 0.4 ,Stuttgart 0.25) is used to distinguish extensive from intensive vegetation and remove these from the green roof vegetation. The intensive vegetation is removed as it was mostly caused by trees growing over the roof, rather than on the roof or by potted plants. Figure 6 also shows the problems associated with merging information from different sources, as the aerial images do not fully compensate for the perspective of oblique sideways acquisition -especially for higher buildings- there is a shift between the building outline and where it is recorded in the image. This can lead to erroneous vegetation along the borders
of buildings, both over and underestimating the vegetation. In Hamburg no near infrared aerial data was available at the time to create the green roof layer.

### 4.4.3   Trees

To resolve tall vegetation (i.e., trees), a range of additional parameters have to be specified that support the generation of leaf area density (see Sect. 5.3). In this study, we aimed at deriving tree height, crown diameter, trunk diameter, tree type and tree
species, as well as the leaf area index, which is described in Sect. 4.4.5. While tree height and crown diameter could be derived from LiDAR data (Fassnacht et al., 2016), the other parameters are very difficult to acquire without extensive field surveys. Luckily, many German cities have tree cadastres, where they store exactly these characteristics to support the maintenance of

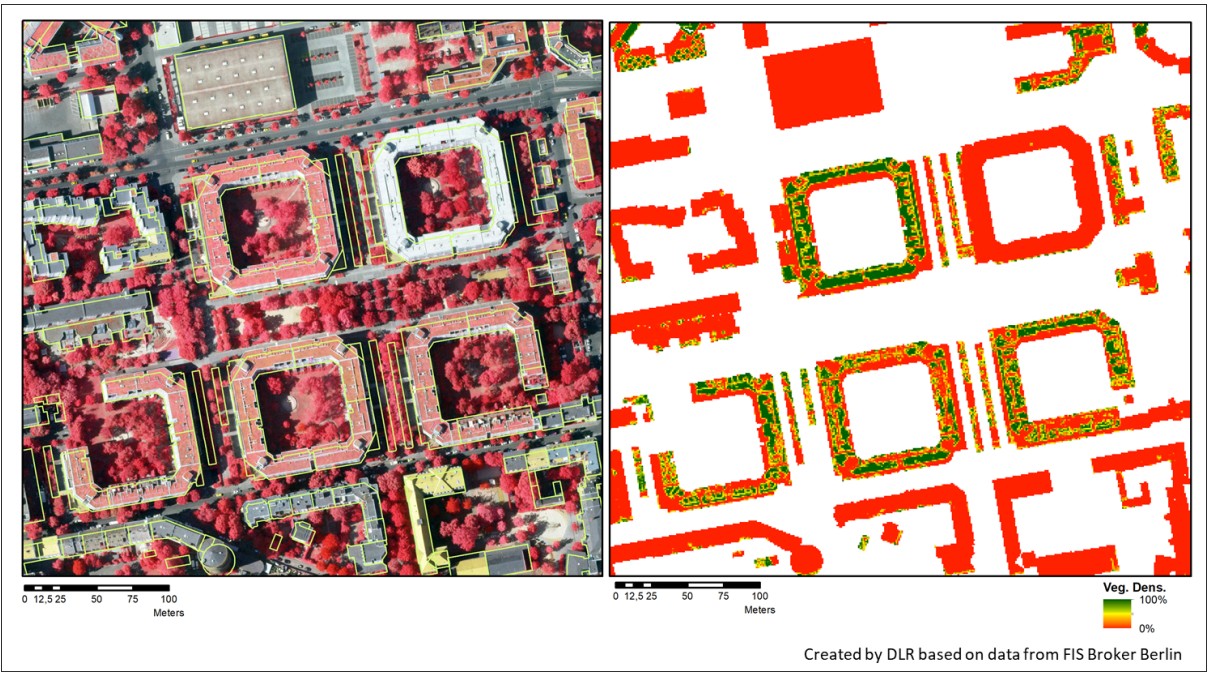

**Figure 6.** 10cm CIR image of a small subset of Berlin on the left, overlaid with the building outlines in yellow. The perspective offset between the orthoimage and the building outlines is especially pronounced for the high round building at the bottom. The resulting vegetation density Map for PALM at 1m resolution is shown in the right illustration.

public trees. Such data sets were available for Berlin, Hamburg and Stuttgart, although the Stuttgart data set only included tree species. Please note that in these municipal data sets only public trees (e.g., along public roads or in parks) are included. Private trees, e.g., in gardens are missing. If no additional sources are available (e.g., as described in Sect. 4.4.4), this means that the uncertainty increases of representing real-world conditions correctly in PALM.

5    To prepare the data for PALM, look-up tables for tree type and species were created, which contain all species and types of trees recorded in the three cities. A class number was assigned to each type and species and then joined to the attribute table of the tree cadastre Shapefile. Varying spellings for the same type or species where taken into account and assigned the same value. For the attributes age, height, trunk diameter and crown size all numbers where checked for plausibility and corrected (typos, wrong unit, shifted columns). The resulting point Shape files were converted to raster (geoTIFF) and then NetCDF. Fig.

10   7 shows exemplary tree type and age maps for a subset of Berlin.

### 4.4.4   Vegetation patch

Instead of providing tree properties of single trees, it is also possible to provide the information on the high vegetation in an area wide manner, as vegetation patches. This is practical, as the tree data sets that where available only cover the public trees. Information on all other trees need to come from another source. Suitable sources are LiDAR data or to some extent also

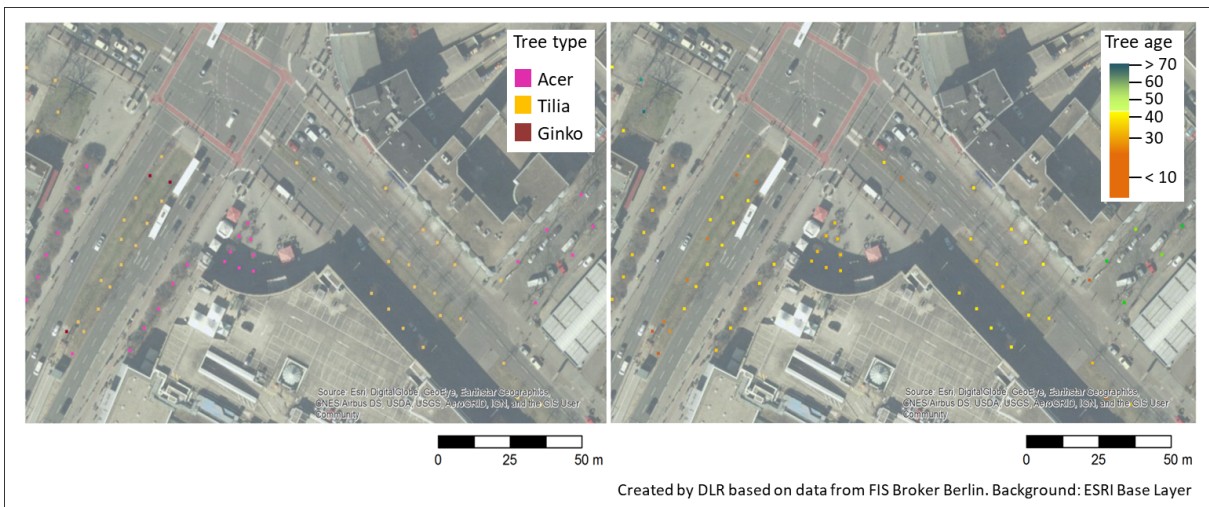

**Figure 7.** Tree species (left) and tree age (right) of municipal trees in Berlin

(governmental) forestry data. Not all cities in Germany provide access to LiDAR data sets, but for Berlin we could use a LiDAR based data set as well as forestry data. The city of Berlin provided a vegetation height map[14], that included the height for all vegetated areas, thus including both public and private trees and shrubs as well as vegetation in parks. For forest vegetation outside Berlin, the Umweltatlas of Berlin (Environmental Atlas) provided information on average tree height, age, type and

trunk diameter at breast height for each forest lot as a Shapefile[15]. The LiDAR based vegetation height and the vegetation height of the Umweltaltas were merged, using the LiDAR based vegetation height as primary data set. The resulting map for the vegetation patch height is shown in Fig.8.

### 4.4.5   Leaf area index

The LAI is an important parameter for the generation of the LAD, which is described in Sect. 5.3. LAI is defined as $m^2$ leaf

area per $m^2$ of ground area and is therefore dimensionless.The LAI varies largely over the phenological cycle. Therefore it is important to approximate the state of the vegetation of the day to be simulated in PALM as best as possible in the input data.Remote sensing is the most suitable data source for representing this temporal variation. Field measurements on the ground only sample single trees, but with area wide remote sensing data an estimation of the LAI of larger area is possible. Typical approaches for LAI estimation make use of vegetation indices in combination with empirical relationships between a

vegetation index and LAI. As only multispectral remote sensing imagery was available for this study, an NDVI based method was selected, making use of Sentinel 2 optical satellite data. Depending on the study area up to three Sentinel 2 image tiles (so called image granules) had to be combined to create a complete coverage of the city area. This is only possible if cloud free image granules of the same or close dates are available. The aim was to create cloud free coverages for each season. Using

---

[14]http://www.stadtentwicklung.berlin.de/umwelt/umweltatlas/i509.htm

[15]http://www.stadtentwicklung.berlin.de/umwelt/umweltatlas/ib504.htm

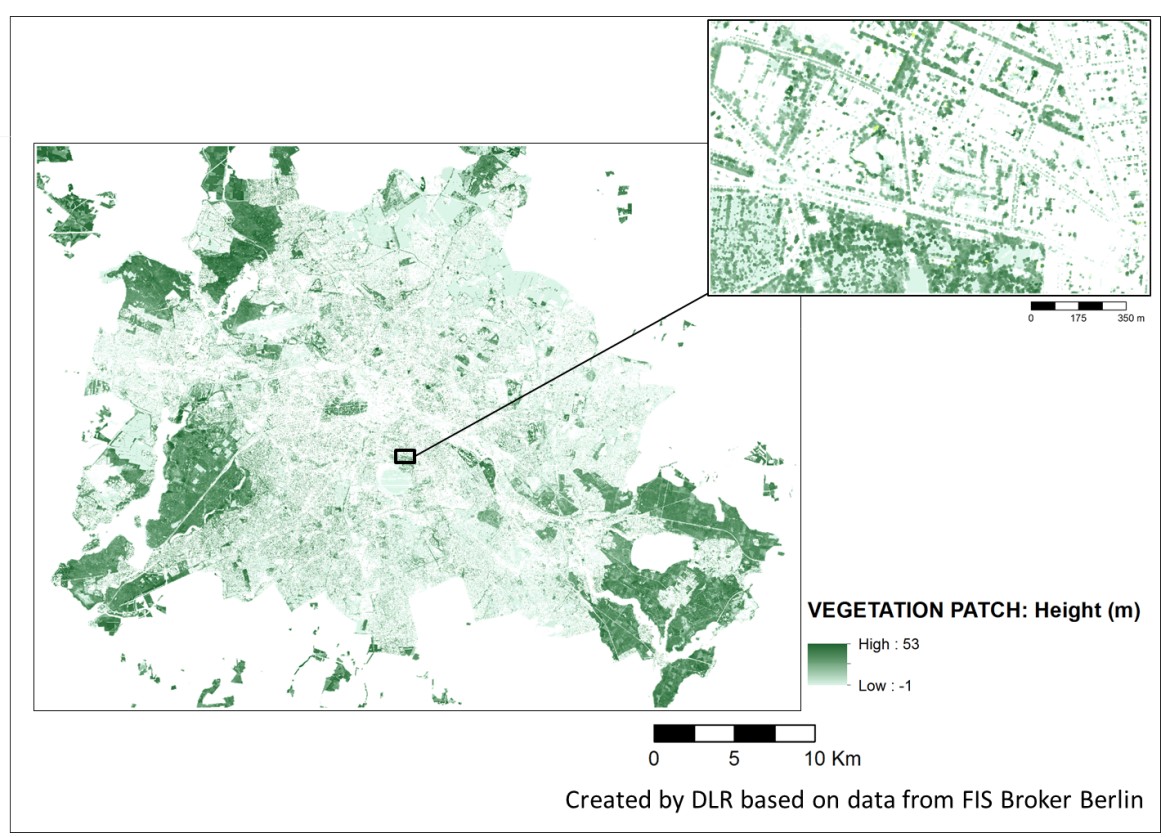

**Figure 8.** Height of vegetation patches in Berlin

data of the year 2017, cloud free images of Berlin, Stuttgart and Hamburg could be created for spring and summer, as well as an additional winter image for Hamburg and an autumn image for Stuttgart.

Using the Timescan processing chain (Esch et al., 2018) the NDVI was derived for all Sentinel 2 scenes. For each date range, an NDVI mosaic image of the study area was created using GDAL tools (GDAL/OGR contributors, 2019). Then the
5  LAI is calculated using an IDL ENVI algorithm (Interactive Data Language and Exelis Visual Information Solutions, Boulder, Colorado) for each study area and date. For this, an empirical relationship between NDVI and LAI is used as documented by Wang et al. (2005) for deciduous forest. All non-vegetation pixels are set to 0 (vegetation mask). As the spatial resolution of Sentinel 2 is 10 m, the required resolution of 1 m is reached by resampling the LAI map using a bilinear resampling method. For a subset of Hamburg the estimated LAI is presented in Fig.9 for spring, summer and winter.

10  **4.5   Pavements**

As a source for the pavement layer airborne hyperspectral would be very useful. Such spatially and spectrally detailed data would allow a differentiated classification of urban surface materials (van der Linden et al., 2019; Roessner et al., 2001).

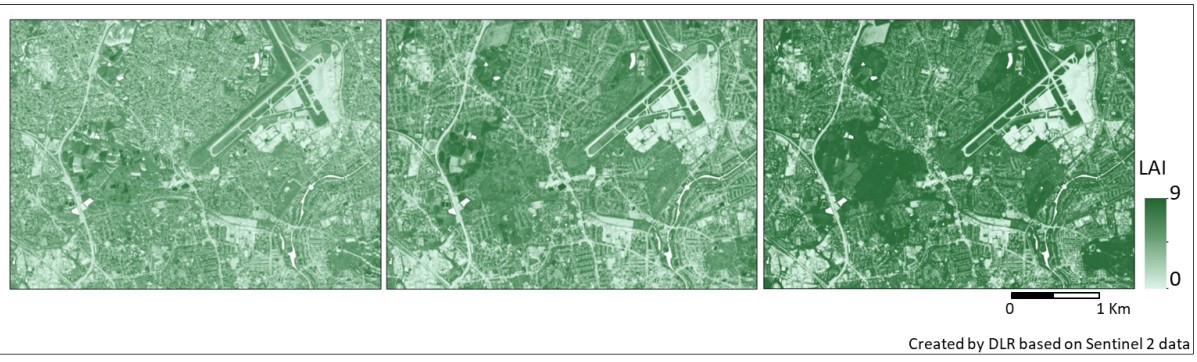

**Figure 9.** LAI for a subset of Hamburg in winter (2-14-2017, left), spring (4-20-2017, center), and summer (8-23-2017, right)

However, due to its experimental nature, hyperspectral data is rarely available for whole cities. Therefore, OSM data was used instead. OSM contributors not only mapped road features, but often also indicated the surface materials. As the contributors do not apply homogeneous labels, a lookup table was created to map all the materials listed in OSM for the test cities to the PALM pavement types. If no surface material is indicated, default materials are assumed for each road type (Table B5). Using another look-up table, the materials were matched to the pavement types listed in the PIDS. As the roads in OSM are line features, each road is buffered with the width or, if not available, a default width for that type of road (Table B5 and B6). After rasterization, the data set is checked for gaps between pavement type, vegetation type, buildings and water. Gaps are filled with the road pavement type by applying a larger buffer (3 x the listed diameter) on the road lines. An example of the resulting pavement type raster map is shown in Fig. 10.

### 4.5.1 Street type and street crossings

For the street types and street crossings data from OSM is used. Street types directly use the classes specified in OSM and are assigned to the road grid cells. If multiple road types cover a pixel, the highest class is assigned. Thus a motorway would have precedence over a primary road ect. A street crossing flag is assigned to all parts of the streets that are marked in OSM as street crossing. As this label is a point feature, all grid cells in a buffer of 15 m around each crossing point are flagged as crossing. At the moment, input data for `street_type` and `street_crossing` is used by PALM's embedded chemistry model (see **?**) to parametrize emissions of chemical compounds during the diurnal cycle.

### 4.6 Water bodies

Multispectral remote sensing is a suitable tool to map water bodies (Ma et al., 2019), but at the high spatial resolution required for building-resolving simulations, the spatial resolution of most satellite data is not sufficient. Also aerial images usually do not provide enough (and calibrated) spectral bands, to distinguish smaller water bodies like fountains or rivulets. Therefore, also in this case OSM was used as primary source for the demo-cities. Unfortunately, it turns out OSM is incomplete regarding water bodies. Therefore the data sets were merged with CLC data for Stuttgart and ALKIS and the Biotope Cataster in Hamburg.

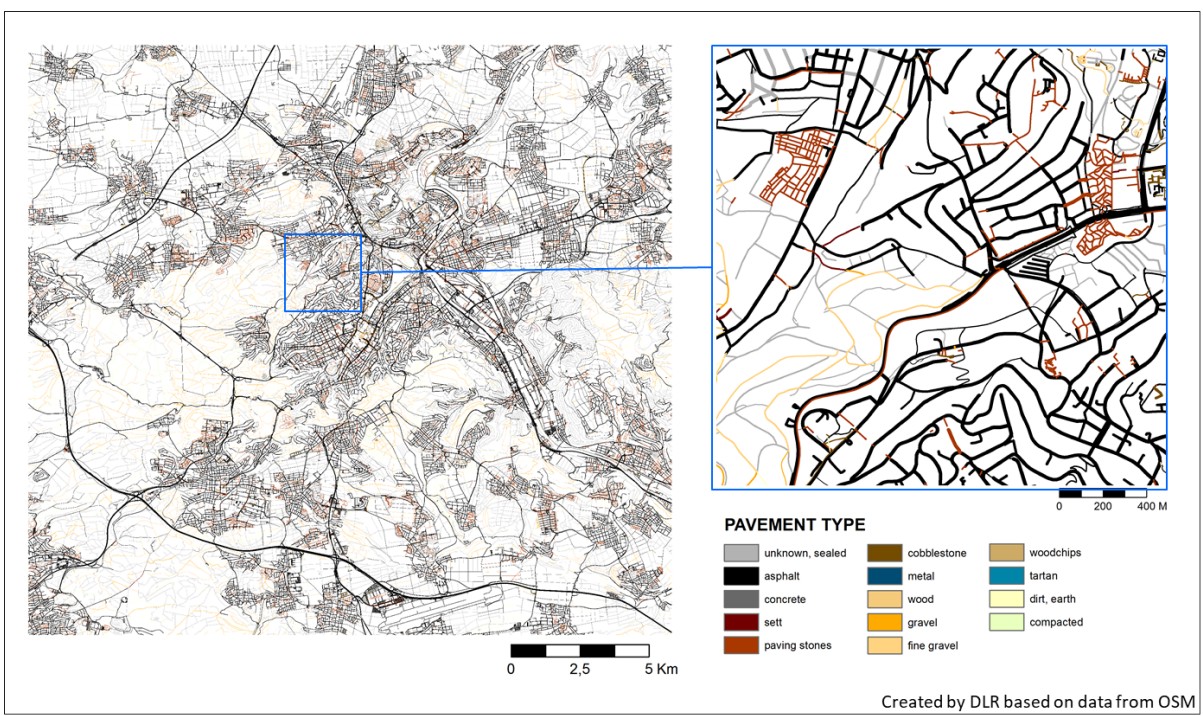

**Figure 10.** Pavement type of Stuttgart

Look-up tables were created to assign a PALM water class to each water feature in the different data sets (see Table B9 and B11). ALKIS polygons are sorted into water types that match the PALM water types (Working Committee of the Surveying Authorities of the States of the Federal Republic of Germany, 2015). Also CLC contains classes that map directly to the PALM water types (European Union, 2017). After the data sets were merged, for Stuttgart and Berlin several important water bodies had to be added manually. The final water type maps for the cities of Stuttgart, Berlin and Hamburg are presented in Fig. 11.

### 4.7 Soils

As soil data is difficult to acquire, especially at resolutions less then 10 m, a horizontally and vertically homogeneous soil type distribution (with soil_type = 1, coarse soil texture) is assumed in this study, i.e. the physical properties of the soil are identical all over the model domain. Further information on the initial state of the soil moisture and temperature at each pixel can be given as LOD0 via Fortran Namelist input, or as LOD1 input given in the dynamic input file (Maronga et al., 2020). The respective soil information can be e.g. take from mesoscale models such as COSMO or WRF, which is described in a separate follow-up paper (Kadasch et al., 2020).

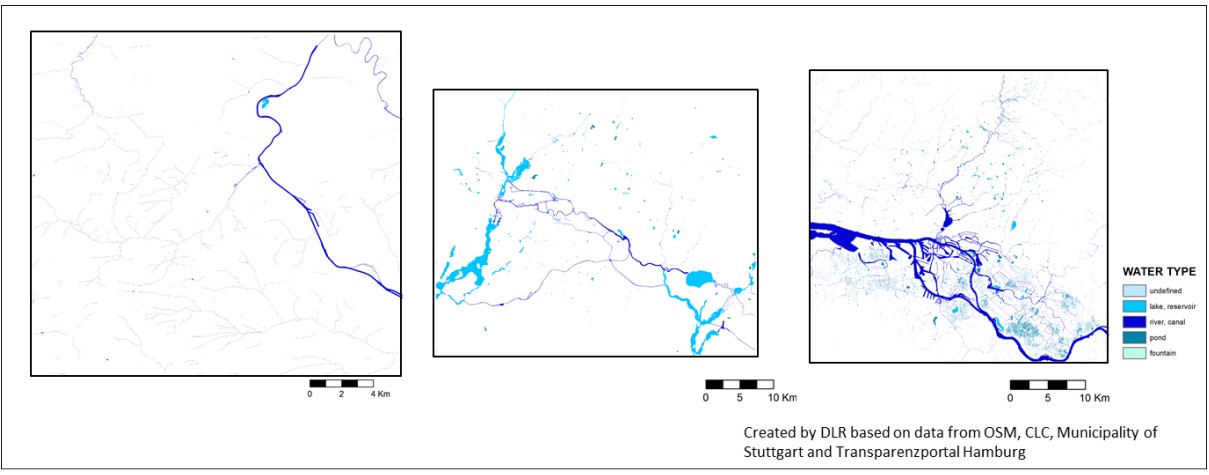

**Figure 11.** Water type of Stuttgart, Berlin and Hamburg (from left to right)

## 5    Preparation of input data for PALM

In this section we discuss PALM static driver generator, the generation of three-dimensional vegetation data in terms of LAD and basal area density (BAD) fields from two-dimensional information as part of the static driver, as well as the internal topography processing which is required to ensure all PALM requirements on the terrain data are met. Note that the netCDF
interface routine in PALM has undergone several improvements since the official release of PALM 6.0. In the following, we will thus describe the status quo for PALM 6.0 in revision 4311.

### 5.1    Using the PALM static driver generator

In order to enable the user to create static drivers for complex scenarios, the Python 3.0-based pre-processing tool `palm_csd` (short for: PALM create static driver) is shipped with PALM. The tool comes with a comprehensive library with netCDF
functions and utility routines that can also easily be plugged into user-specific Python codes, and which take care of the correct formatting of static driver files that comply with PALM's netCDF interface. `palm_csd` itself, however is a wrapping and compiling tool, which compiles static drivers based on already processed and rasterized geospatial data in netCDF format, but it cannot process other geospatial file formats (e.g., GeoTIFF or Shapefile). At the moment, it is thus up to the user to process such data manually and provide `palm_csd` with PIDS conform NetCDFs. Currently, input data for `palm_csd` is available
for the cities of Berlin, Hamburg, and Stuttgart in Germany, for which input data was processed based on the data sources outlined in Sect. 4, but the user is free to provide his own data to be processed by `palm_csd`. Note that while data for Berlin and Hamburg is freely available for the general public, data of Stuttgart is restricted to be used within the [UC][2] project. During the pre-processing of the data for Berlin, Hamburg and Stuttgart it was aimed to automate the pre-processing steps as much as possible by implementing the geo-processing in scripts and reduce manual processing in GIS software. In the next phase

of [UC]$^2$ it is planned to develop a pre-processing tool that will support users to generate the input data in PALM conform formats.

palm_csd is steered via a configuration file in which input files, basic settings, and default values are defined. Once this configuration file is set-up, the user can generate their own static driver files that include correct metadata and possibly geo-referencing (depending on suitable input data) for PALM and that will also be written to PALM's output data for post-processing and visualization.

We plan to extent palm_csd for generic and academic setups as well as with a graphical user interface in near future. Moreover, we plan to implement a comprehensive checking routine so ensure compatibility with PALM, which is currently done within PALM itself.

## 5.2 Internal topography processing

During the initialization of PALM, the provided topography data, encompassing terrain height and buildings, is further processed and could possibly be slightly modified, e.g., to fulfill numerical requirements or to reduce the use of computational resources.

The model surface in PALM is internally defined at $z = 0\,\mathrm{m}$. Therefore, in a first step, PALM internally computes the relative terrain height $z'_\mathrm{t} = z_\mathrm{t} - z_\mathrm{t,min}$, where $z_\mathrm{t,min}$ is the minimum terrain height occurring within the model domain. Thus, the minimum $z'_\mathrm{t}$ coincides with the model surface at $z = 0\,\mathrm{m}$ and the first vertical grid level has at least one grid point that lies within the atmosphere. For instance, if $z_\mathrm{t}$ is given in meters above sea level and we would use this without any further processing, many grid points may lie below the Earth surface, being a waste of computational resources without providing any additional value. In case of a nested simulation setup with a root domain and various child domains, $z_\mathrm{t,min}$ is calculated as the minimum terrain height over all domains, in order to have the same reference height for all model domains and avoid artificially induced elevation changes at the domain borders between the parent and the child models.

In the following, $z'_\mathrm{t}(y,x)$ is projected onto the discrete grid, while all grid points are flagged as terrain that are located below $z'_\mathrm{t}(y,x)$, as illustrated in Fig. 12 by the dashed black line.

In a second step buildings are mapped on top of the discrete terrain, which is illustrated schematically in Fig. 12. Especially when the underlying terrain is not flat but elevation changes occur below a building, roof shapes should be maintained, so that buildings can't be simply mapped on top of $z'_\mathrm{t}$. Hence, the underlying terrain below a single building (which is identified by its building_id) is padded up to the level of the highest $z'_\mathrm{t}(y,x)$ within the building-covered area with respective building_id : $z'_\mathrm{t}(y,x) = max(z'_\mathrm{t}(y,x))_{ID}$, i.e., the terrain below the building is flattened (please see the hashed areas in Fig. 12). This guarantees that building and roof shapes are maintained even at steep slopes. However, an exception is made for bridges (identified by building_type = 7) where buildings_3d is directly mapped on top of $z'_\mathrm{t}$. Flatting the terrain below the bridge to the highest terrain height (often the top of the levee) would otherwise introduce barrier-like topography structures.

While buildings are mapped onto the terrain, grid points that lie within buildings and below terrain are internally flagged, in order to classify building- or land-surfaces during the surface initialization (see Sect. 2.4). The padded grid points below

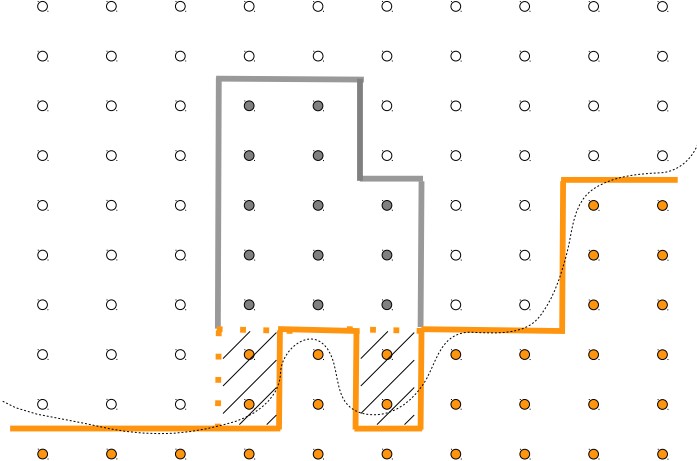

**Figure 12.** Schematic illustration on how buildings are mapped onto the underlying terrain. The thin dashed black line indicates the original relative terrain height $z'_\mathrm{t}$. Solid orange lines indicate the original discrete terrain surface, while dashed orange lines indicate the resulting discrete terrain surface after the terrain is flattened below buildings, as indicated by the hashed areas. Grey solid lines indicate building surfaces. Orange and grey coloured points indicate terrain and building grid points, respectively, while non-filled points indicate atmospheric grid points.

buildings will be not flagged as building but as land-surfaces, while these artificially introduced vertical land-surfaces will be initialized using the given `vegetation_type` or `pavement_type` at the adjacent grid cell.

After the topography is projected onto the discrete grid, it may contain single pixel cavities or chimney-like holes that are only resolved by one grid point. Due to numerical issues, such one-grid-point cavities must be filtered. In many cases these filtered cavities are building courtyards that are resolved by only one grid point. In this case, the courtyard grid point, which might be originally given, e.g., a `vegetation_type`, is internally flagged and re-set to a building grid point while it obtains `building_type`, `building_id` and, if available, `building_pars` from the nearby building grid point. Hence, we filter such one-grid point cavities during the model initialization, meaning that small differences might occur between the final building and terrain geometry in the model and the provided one in the static driver.

## 5.3 Generation of three-dimensional leaf area density and basal area density fields

When using PALM at very high resolution in the order of 1 m, vegetation like tall shrubs or trees can not be represented by common parameterizations that assume the vegetation canopy to be flat and represented, e.g., by a roughness length. Under such conditions, PALM employs a plant canopy model in which high vegetation can be represented in terms of three-dimensional LAD fields. As geospatial data usually does not yield any exact three-dimensional information, three-dimensional LAD and BAD fields must be estimated from two-dimensional data and other data sources. In order to allow for a pseudo-automated generation of LAD and BAD fields, `palm_csd` comes with two different routines for creating vegetation canopies: A routine for single trees as often found in urban environments, and a routine for creating vegetation canopies like forests and parks. In the following we will outline the basics of both routines. Note, however, that both routines are still in experimental stage and will be further developed and evaluated in the near future. In the following we will thus describe the status quo of these routines.

### 5.3.1 Generation of leaf area density and basal area density fields for single trees

Single trees, whose growth is seldom affected by other trees or obstacles can be characterized in terms of three-dimensional LAD and BAD fields by a limited number of parameters. In `palm_csd` these are the maximum tree height, crown diameter, crown shape, trunk diameter, height of the maximum LAD value, and the aspect ratio of tree crown diameter to tree crown height (see Figs. 13 and 14). In German cities, several of these parameters are available from tree cadastral register data. For example, for Berlin more than 400 000 municipal trees are collected in a publicly available database including information about tree species, tree height, crown diameter, stand age, and trunk diameter.

The single-tree canopy generator in `palm_csd` is called for each individual tree and the following information is passed to the generator: location $(y, x)$ of the tree centre, tree type (i.e., genus), tree height, LAI, crown diameter, and trunk diameter at breast height. If one or more of these parameters is not provided, a default value from a look-up table (see Table 5) is used, which was generated based on averaging each tree parameter for each tree type in the Berlin tree database. This look-up table also includes default values for the tree shape, the ratio between crown height to crown width, LAI values for summer and winter time, and the height of the LAD maximum. Note for some of the latter parameters only dummy values are currently available (crown height to width ratio, LAI, height of LAD maximum) and more effort will be needed to fill this table with reasonable data. The tree generator allows for six different tree shapes which are shown in Fig. 14 and which cover most

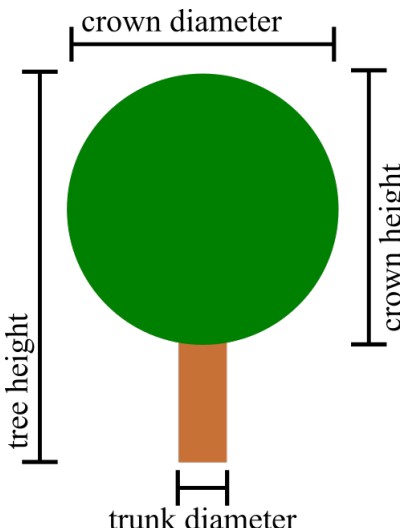

**Figure 13.** Schematic view of the parameters for a spherical tree shape through which the three-dimensional structure of trees are constructed in the single-tree canopy generator.

of the commonly observed shapes for single trees. The generation of a three-dimensional LAD volume then consists of two steps. First, the volume covered with leaves is determined based on the shape, crown diameter, tree height, and the ratio of crown height to width. Second, the three-dimensional LAD field is created using an exponentially increasing LAD towards the outer shell of the foliage. This approach is based on the empirical finding that sun light is absorbed when entering the foliage resulting in decreasing production of leaves.

Calculation of three-dimensional BAD fields is available using an interim solution, where the BAD field is calculated from the given trunk diameter, which is taken as constant up to the center of the tree crown. At the moment, the canopy generator only allows to treat each grid volume as either (impermeable) stem or no stem. The representation of grid volumes partially covered by trunks is thus not possible at the moment. BAD values within the crown canopy is calculated as

$$BAD = 0.1 \cdot (1 - LAD), \tag{2}$$

which reflects increasing BAD towards the center of the crown. An example of both lad and bad fields for an idealized spherical-shaped tree is shown in Fig. 15. Note, however, that PALM currently only supports LAD so that only the foliage is read from the static driver. The import of BAD data will be realized in near future.

### 5.3.2 Generation of leaf area density fields for tree stands

In many cases, information on individual trees is not available or tree stands (e.g., forests) have to be represented as a three-dimensional canopy. This is commonly realized by treating each column $(y, x)$ separately and using normalized LAD profiles that are representative for homogeneous canopies. In `palm_csd` the method of Markkanen et al. (2003) based on a vertical LAD distribution that is derived from a given LAI field as well as two parameters $\alpha$ and $\beta$, which can be varied by the user to

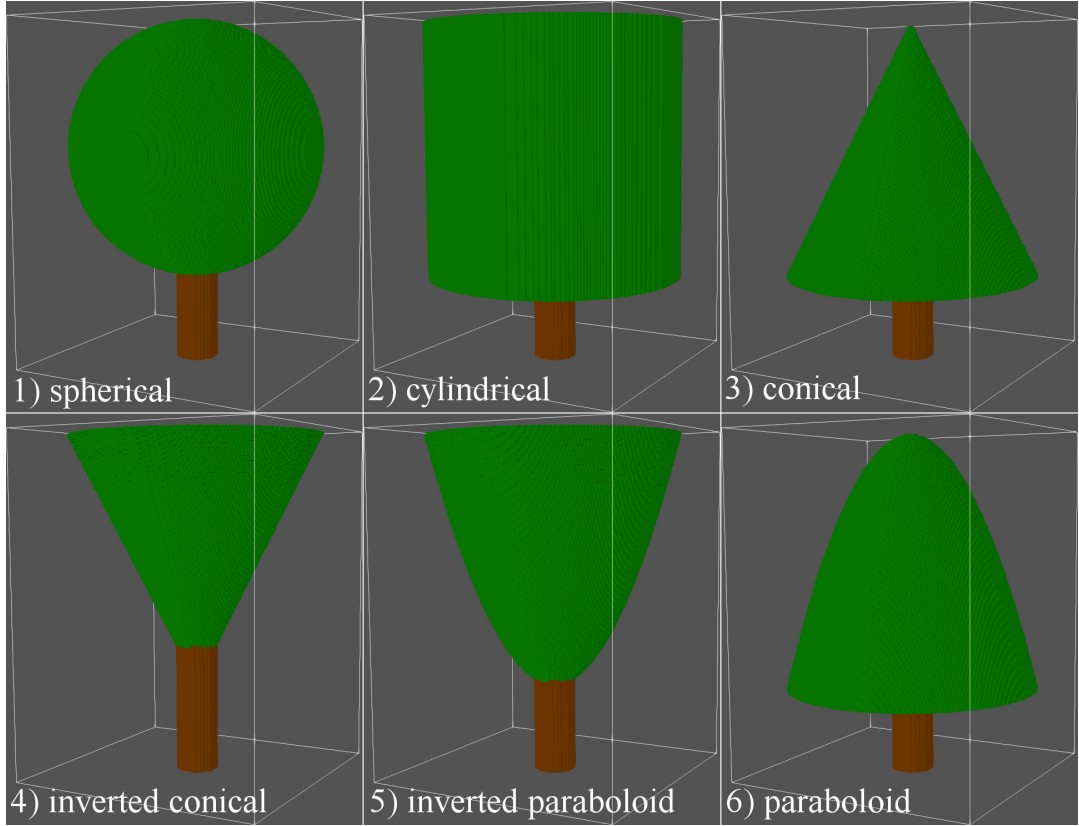

**Figure 14.** Overview of tree shapes available in the single-tree canopy generator. Green surfaces represent the foliage while brown surfaces represent the tree trunk. The shown sketches were created for a raster size of 0.05 m using a tree height of 8 m, a crown width of 6 m, a crown height to width ratio of 1, and a trunk diameter of 1 m.

represent different types of tree stands. Additionally, a two-dimensional vegetation height field can be prescribed (if available) in order to take into account varying tree heights within the canopy stand. If information on LAI and vegetation height is not available, the user has to provide default values instead. Using this method it is possible to generate idealized vegetation canopies in terms of LAD fields, but it provides no means to derive BAD information. In the future we plan to use a similar method as described in Bohrer et al. (2007) to create BAD fields by synthetically localizing tree trunks.

## 6 Example for a real-world application

To demonstrate the suitability of the input data prepared as described above, the following shows an example static driver and simulation of a part of Berlin. The static driver of this example can be found in the supplement to this paper.

Figure 16 shows static input data for a nested simulation with 1-m grid resolution. The simulation setup contains several residential and office buildings with different heights around the Ernst-Reuter Platz, Berlin. The streets indicate different type

**Table 5.** List and average properties of the most common street trees in Berlin.

| Genus | Quantity | Tree height | Crown diameter | Trunk diameter (DBH) | Age |
|---|---|---|---|---|---|
| Unit | | m | m | m | yr |
| Acer | 119863 | 12.1 | 7.1 | 1.0 | 41.8 |
| Aesculus | 24601 | 12.2 | 7.0 | 1.3 | 51.3 |
| Ailanthus | 1833 | 14.2 | 8.5 | 1.3 | 46.4 |
| Alnus | 3764 | 16.3 | 5.9 | 1.3 | 47.9 |
| Betula | 25580 | 14.0 | 6.0 | 1.0 | 39.7 |
| Carpinus | 15862 | 10.2 | 6.0 | 0.8 | 33.9 |
| Corylus | 7789 | 8.8 | 4.9 | 0.7 | 29.6 |
| Crataegus | 8570 | 5.9 | 3.5 | 0.5 | 26.2 |
| Fagus | 9923 | 17.5 | 9.9 | 1.7 | 91.3 |
| Fraxinus | 14217 | 10.7 | 5.6 | 0.9 | 35.1 |
| Ginkgo | 1289 | 8.4 | 4.0 | 0.6 | 24.8 |
| Gleditsia | 2746 | 10.6 | 6.3 | 0.8 | 29.3 |
| Juglans | 1124 | 9.5 | 6.8 | 0.9 | 37.2 |
| Larix | 1218 | 16.7 | 7.1 | 1.2 | 51.3 |
| Malus | 3146 | 4.8 | 3.5 | 0.5 | 21.5 |
| Picea | 3122 | 13.6 | 3.6 | 0.9 | 37.5 |
| Pinus | 13985 | 15.7 | 6.3 | 1.2 | 49.2 |
| Platanus | 25548 | 14.7 | 9.9 | 1.4 | 50.6 |
| Populus | 16750 | 20.6 | 8.8 | 1.7 | 54.9 |
| Prunus | 14594 | 7.0 | 4.9 | 0.7 | 27.8 |
| Pseudotsuga | 2652 | 17.6 | 5.8 | 1.2 | 42.8 |
| Pyrus | 2779 | 6.0 | 2.6 | 0.4 | 19.9 |
| Quercus | 58416 | 14.2 | 8.0 | 1.3 | 55.6 |
| Robinia | 20341 | 13.9 | 6.7 | 1.1 | 42.4 |
| Salix | 4891 | 13.8 | 7.9 | 1.6 | 44.5 |
| Sophora | 1852 | 10.2 | 7.7 | 1.0 | 36.9 |
| Sorbus | 8251 | 6.7 | 4.0 | 0.6 | 28.2 |
| Taxus | 2767 | 7.6 | 5.2 | 0.8 | 43.3 |
| Tilia | 156496 | 12.6 | 7.2 | 1.1 | 45.7 |
| Ulmus | 8729 | 13.8 | 7.4 | 1.2 | 42.2 |

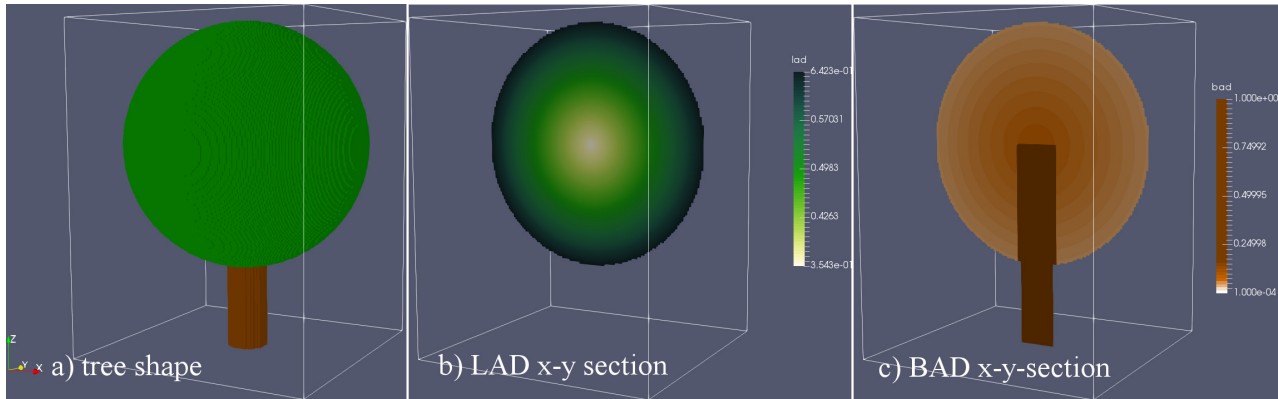

**Figure 15.** Exemplary distribution of LAD and BAD fields for a spherical-shaped tree. Shown are (a) the three-dimensional tree surface and $x - y$-sections of (b) LAD and (c) BAD through the center of the tree.

of pavement, with e.g. asphalt, concrete and cobblestones. Further, grass areas and evergreen shrubs are present within the simulation domain, as well as water ponds and the Spree river. The LAI indicates areas with resolved-scale trees, with single trees planted along pedestrian walks (e.g. in the northwest area of the displayed domain) but also more continuous areas with trees within the Tiergarten park area (see southeast area of the displayed domain).

Figure 17 shows the corresponding horizontal cross-sections of surface net radiation, surface temperature, as well as 2-m potential temperature for an area around Ernst-Reuter Platz in Berlin, Germany. The surface net radiation and surface temperature show a clear dependence on the underlying surface, e.g. the building roofs indicate a larger negative surface net radiation but a higher surface temperature compared to other surfaces. Further, pavement surfaces indicates a only small negative surface net radiation but relatively low surface temperatures. The river and the small ponds show a comparably high surface temperature resulting in a larger negative surface net radiation. The 2-m potential temperature correlates with the underlying surface as well, with e.g. lower values within wide street canyons and larger values in densely built-up areas.

The input data described in this manuscript is also already used in several other studies. A validation of the dynamic core of PALM against wind-tunnel data for a city quarter in Hamburg, Germany, is presented by Gronemeier et al. (2020), where the input data of Hamburg as described in this paper was used. A subset of 6000 m by 2880 m around the HafenCity has been selected and rotated counter-clockwise by 200 degrees to match the prevailing wind direction. Comparison between the PALM simulation result and the wind tunnel experiment show mainly similar wind directions, but lower wind speeds. As this difference in wind speed decreases at lower spatial resolutions, Gronemeier et al. (2020) assume that this is caused by an overestimated $z0$ of the building walls in the PALM simulation. The roughness of walls is further increased when building walls are not aligned to the grid and 'stair case'- like walls occur. This issue cannot be avoided all together but is minimised when using small grid cells. Furthermore, Gronemeier et al. (2020) emphasised that it is of utmost importance to cautiously check the input data. Especially in large data sets, erroneous or false buildings are hard to spot, but can have large influences on the wind pattern, especially if they are located in an up-wind part of the study area.

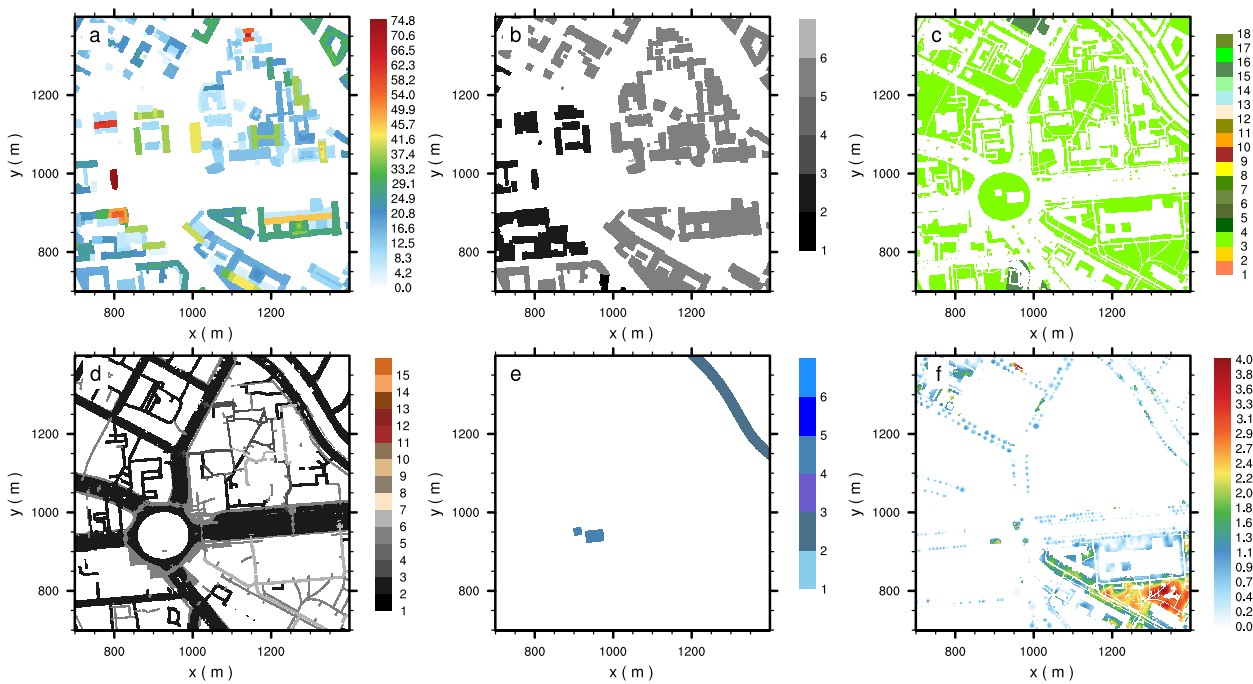

**Figure 16.** $x-y$-cross-section of static input data for a nested simulation with 1-m grid resolution for a winter scenario for an area around Ernst-Reuter Platz in Berlin, Germany: (a) building height, (b) building type, (c) vegetation type, (d) pavement type, (e) water type, (f) leaf-area index of resolved vegetation (trees). For the sake of illustration (f) displays the leaf-area index instead of the leaf area density, with the leaf-area index being the vertically integrated leaf-area density. Please note, in (b)–(e) the lable bar contains all possible `*_type` variables as indicated in Tab. A1, A2, A3, and A5, though only a few types are defined within the displayed area. (b) displays `building_type` 1 (R1), 2 (R2) and 5 (O2). (c) displays `vegetation_type` 3 (short grass) and 15 (evergreen shrubs). (d) displays `pavement_type` 2 (asphalt), 3 (concrete), 4 (sett), 5 (paving stone) and 6 (cobblestone). (e) displays `water_type` 2 (river) and 4 (pond).

Salim et al. (2020) assessed the importance of radiative transfer processes in urban climate models, in this case PALM6.0. They analysed both a simple urban configuration as well as a real world configuration of the area around the Ernst-Reuter-Platz in Berlin using the same data as used in the example above. They found comparable results regarding the radiative transfer processes for the simple urban area as for the real world area, although the results for the real world area were much more heterogenious, as to be expected.

The sensitivity of the simulation results with respect to variations in the input data is further discussed in Belda et al. (2020) (using a different set of input data). In this study, the sensitivity of air and surface temperature, MRT, PET and PM10 within the PALM6.0 as a response to modification of basic surface material parameters as well as to common urbanistic strategies was evaluated. It was found, that for this kind of simulations, the albedo and emissivity as well as thermal conductivity of walls and volumetric heat capacity of the materials play an important role (Belda et al., 2020).

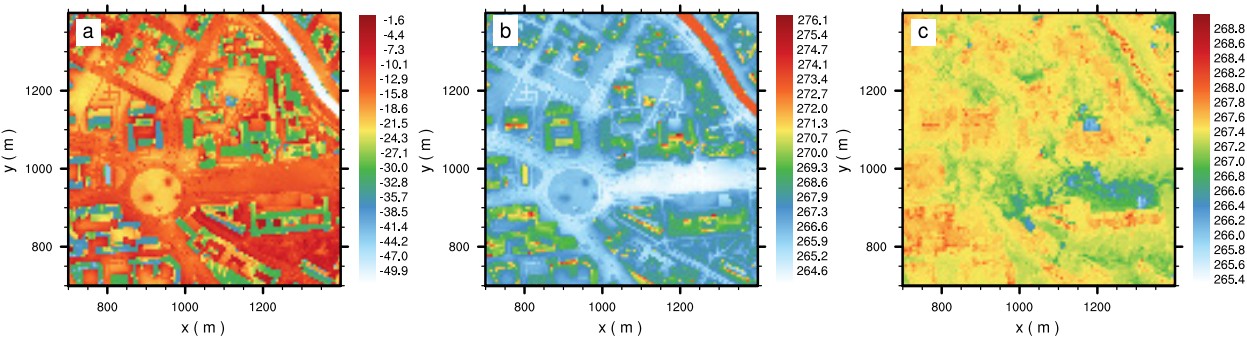

**Figure 17.** $x - y$-cross-section of (a) surface net radiation $(\mathrm{W\,m^{-2}})$, (b) surface temperature (K), and (c) 2-m potential temperature (K) for a nested simulation with 1-m grid resolution for a winter scenario at 18.01.2017, 00:00 UTC, for an area around Ernst-Reuter Platz in Berlin, Germany. The displayed area is the same as in 16.

These experiences with the input data indicate, that on the one hand the correct identification of urban objects and heights is important, but on the other hand also the characterization of these objects as currently defined in the building and vegetation parameters at LOD1 and higher.

Furthermore, a thorough validation of PALM in a real urban environment against in-situ measurements is presented in Resler
et al. (2020). They especially focus on the representation of surface temperatures in PALM at different type of urban surfaces and also emphasize the need of accurate input data to be in agreement with in-situ measurements. A dedicated evaluation of the PALM 6.0 model system for the city of Berlin and Stuttgart is currently on its way within the project framework of $[\mathrm{UC}]^2$ (Scherer et al., 2019).

## 7 Conclusions

In the previous sections, the input data requirements of PALM are described, it is demonstrated how this data can be prepared and what steps are carried out in `palm_csd` to set-up the static driver to have all input data ready for PALM. PALM comes with a framework that enables micro climate simulations for a real-world urban environment. Different levels of detail can be provided to PALM. If the model is run with interactive building- and land surfaces, a minimum of seven spatial parameters is required: soil type, building height, building id, building type, vegetation type, pavement type and water type. Each of these
parameters can optionally be specified in more detail, based on available data. Sect. 3 and 4 illustrate that a vast amount of data exists, but rarely exactly in the format required by urban micro climate models. Exemplary for this is the building type. The combination of building use and building age yields the PALM building types. However, it needs to be further analysed if this is the most accurate representation of the energetic properties of the building, as the building age often doesn't include any information on renovation and modernisation of the building which can have a huge effect on the energetic properties of a
building. Selecting and acquiring suitable data sets is a major task, that should be weighted against the available resources to

pre-process the input data and the desired detail of the PALM simulations. Additionally, the varying quality of different data sources result in different uncertainties of the input parameters. There are uncertainties resulting from the spatial resolution (e.g., the ability to distinguish small objects), but there can also be mapping or labeling errors and omissions. How these uncertainties propagate into the simulation results needs to be investigated in more detail. To support users in the decision,

which parameter are worth the effort of acquiring and preparing more detailed information, sensitivity analyses of the input data sets are planned. Also additional evaluation of LOD1 parameters (such as albedo or thermal conductivity) need to be carried out, as some of them have a large influence on the simulation result. For albedo, it could also be considered to derive such values separately for each grid cell, e.g. using remote sensing.

As the pre-processing of the input data is tedious, it is aimed to develop a processing chain that support users in format-
10 ting their GIS data (e.g., Shapefiles,geoTIFF, WFS ect.) into a NetCDF file following the requirements of PALM. Existing standardized workflows such as those of WUDAPT (Ching et al., 2018) or MApUCE (Bocher et al., 2018) are interesting examples that should be considered following. To support model users in their data acquisition, a data base with freely available geospatial data for the mandatory set of parameters of PALM is aimed at. This will provide users a starting point for running PALM simulations. The primarily target will be Germany but also Europe wide or global data can be included, as far as the
15 data sources allow this.

## 8 Code availability

The PALM model system is distributed under the GNU General Public License v3 (http://www.gnu.org/copyleft/gpl.html). The model source, documentation, user manual, and online tutorial are freely-available and can be downloaded from http://palm-model.org. The pre-processing tool `palm_csd` to prepare and create a PALM Static driver is shipped with PALM and
20 is available under https://doi.org/10.25835/0041607.

## 9 Data availability

In the supplements, a sample static driver is available for a small area in Berlin near Ernst-Reuter Platz, Germany, with 1 m spatial resolution. The static driver is prepared for a winter scenario with leafless deciduous trees. The model domain is $256 \times 256 \, \mathrm{m}^2$ in the horizontal directions.

**Appendix A: Palm Input Data Standard (PIDS) tables**

**Table A1.** Land use classification parameters according to vegetation_type based on the Integrated Forecasting System (IFS) classification. Note that the land use class 13 (ice caps and glaciers) has not been implemented yet. $r_{c,min}$ is the minimum canopy resistance, $LAI$ is leaf area index, $c_{veg}$ is vegetation coverage, $g_D$ is a canopy resistance coefficient, $z_0$ and $z_{0,h}$ are the roughness lengths for momentum and heat, respectively, $\Lambda_s$ and $\Lambda_u$ are the bulk heat conductivities between skin and soil layer for stable and unstable stratification, respectively, $f_{sw,in}$ is the fraction of absorbed shortwave radiation by the vegetation canopy, and $\epsilon$ is the surface emissivity.

| Description | Class | $r_{c,min}$ | $LAI$ | $c_{veg}$ | $g_D$ | $z_0$ | $z_{0,h}$ | $\Lambda_s$ | $\Lambda_u$ | $f_{sw,in}$ | albedo_type | $\epsilon$ |
|---|---|---|---|---|---|---|---|---|---|---|---|---|
| | | s m$^{-1}$ | m$^2$ m$^{-2}$ | | hPa$^{-1}$ | m | m | W m$^{-2}$ K$^{-1}$ | W m$^{-2}$ K$^{-1}$ | | | |
| bare soil | 1 | 0 | 0.0 | 0.0 | 0.0 | 0.005 | $0.5\cdot10^{-4}$ | 0.0 | 0.0 | 0.0 | 17 | 0.94 |
| crops, mixed farming | 2 | 180 | 3.0 | 1.0 | 0.0 | 0.1 | $1.0\cdot10^{-4}$ | 10.0 | 10.0 | 0.05 | 2 | 0.95 |
| short grass | 3 | 110 | 2.0 | 1.0 | 0.0 | 0.03 | $0.3\cdot10^{-4}$ | 10.0 | 10.0 | 0.05 | 2 | 0.95 |
| evergreen needleleaf trees | 4 | 500 | 5.0 | 1.0 | 0.03 | 2.0 | 2.0 | 20.0 | 15.0 | 0.03 | 5 | 0.97 |
| deciduous needleleaf trees | 5 | 500 | 5.0 | 1.0 | 0.03 | 2.0 | 2.0 | 20.0 | 15.0 | 0.03 | 6 | 0.97 |
| evergreen broadleaf trees | 6 | 175 | 5.0 | 1.0 | 0.03 | 2.0 | 2.0 | 20.0 | 15.0 | 0.03 | 8 | 0.97 |
| deciduous broadleaf trees | 7 | 240 | 6.0 | 0.99 | 0.13 | 2.0 | 2.0 | 20.0 | 15.0 | 0.03 | 9 | 0.97 |
| tall grass | 8 | 100 | 2.0 | 0.7 | 0.0 | 0.47 | $0.47\cdot10^{-2}$ | 10.0 | 10.0 | 0.05 | 8 | 0.97 |
| desert | 9 | 250 | 0.05 | 0.0 | 0.0 | 0.013 | $0.013\cdot10^{-2}$ | 15.0 | 15.0 | 0.0 | 3 | 0.94 |
| tundra | 10 | 80 | 1.0 | 0.5 | 0.0 | 0.034 | $0.034\cdot10^{-2}$ | 10.0 | 10.0 | 0.05 | 11 | 0.97 |
| irrigated crops | 11 | 180 | 3.0 | 1.0 | 0.0 | 0.5 | $0.5\cdot10^{-2}$ | 10.0 | 10.0 | 0.05 | 13 | 0.97 |
| semidesert | 12 | 150 | 0.5 | 0.1 | 0.0 | 0.17 | $0.17\cdot10^{-2}$ | 10.0 | 10.0 | 0.05 | 2 | 0.97 |
| ice caps and glaciers (*) | 13 | 0 | 0.0 | 0.0 | 0.0 | 1.3E-3 | $1.3E\cdot10^{-4}$ | 58.0 | 58.0 | 0.00 | 14 | 0.97 |
| bogs and marshes | 14 | 240 | 4.0 | 0.6 | 0.0 | 0.83 | $1.3\cdot10^{-4}$ | 10.0 | 10.0 | 0.05 | 4 | 0.97 |
| evergreen shrubs | 15 | 225 | 3.0 | 0.5 | 0.0 | 0.10 | $0.1\cdot10^{-2}$ | 10.0 | 10.0 | 0.05 | 4 | 0.97 |
| deciduous shrubs | 16 | 225 | 1.5 | 0.5 | 0.0 | 0.25 | $0.25\cdot10^{-2}$ | 10.0 | 10.0 | 0.05 | 4 | 0.97 |
| mixed forest/woodland | 17 | 250 | 5.0 | 1.0 | 0.03 | 2.0 | 2.0 | 20.0 | 15.0 | 0.03 | 7 | 0.97 |
| interrupted forest | 18 | 175 | 2.5 | 1.0 | 0.03 | 1.1 | 1.1 | 20.0 | 15.0 | 0.03 | 8 | 0.97 |

**Table A2.** Building classification parameters according to `building_type` based on building age and usage.

| Description | Class | Year of construction | Use |
|---|---|---|---|
| R1 | 1 | $< 1951$ | residential |
| R2 | 2 | $1951 - 2000$ | residential |
| R3 | 3 | $> 2000$ | residential |
| O1 | 4 | $< 1951$ | office |
| O2 | 5 | $1951 - 2000$ | office |
| O3 | 6 | $> 2000$ | office |

**Table A3.** Pavement classification parameters according to `pavement_type` based on OpenStreetMaps. Thermal conductivity and heat capacity settings of the sub-surface pavement layers are given in Tab. A4. Underlined values are preliminary. $z_0$ and $z_{0,h}$ are the roughness lengths for momentum and heat, respectively, and $\epsilon$ is the surface emissivity.

| Description | Class | $z_0$ | $z_{0,h}$ | $\epsilon$ | `albedo_type` |
|---|---|---|---|---|---|
| | | m | m | | |
| asphalt/concrete mix | 1 | $5.0{\cdot}10^{-2}$ | $5.0{\cdot}10^{-4}$ | 0.97 | 18 |
| asphalt (asphalt concrete) | 2 | $5.0{\cdot}10^{-2}$ | $5.0{\cdot}10^{-4}$ | 0.94 | 19 |
| concrete (Portland concrete) | 3 | $1.0{\cdot}10^{-2}$ | $1.0{\cdot}10^{-4}$ | 0.98 | 20 |
| sett | 4 | $1.0{\cdot}10^{-2}$ | $1.0{\cdot}10^{-4}$ | 0.93 | 21 |
| paving stones | 5 | $1.0{\cdot}10^{-2}$ | $1.0{\cdot}10^{-4}$ | 0.97 | 22 |
| cobblestone | 6 | $1.0{\cdot}10^{-2}$ | $1.0{\cdot}10^{-4}$ | 0.97 | 23 |
| metal | 7 | $1.0{\cdot}10^{-2}$ | $1.0{\cdot}10^{-4}$ | 0.97 | 24 |
| wood | 8 | $1.0{\cdot}10^{-2}$ | $1.0{\cdot}10^{-4}$ | 0.94 | 25 |
| gravel | 9 | $1.0{\cdot}10^{-2}$ | $1.0{\cdot}10^{-4}$ | 0.98 | 26 |
| fine gravel | 10 | $1.0{\cdot}10^{-2}$ | $1.0{\cdot}10^{-4}$ | 0.93 | 27 |
| pebblestone | 11 | $1.0{\cdot}10^{-2}$ | $1.0{\cdot}10^{-4}$ | 0.97 | 28 |
| woodchips | 12 | $1.0{\cdot}10^{-2}$ | $1.0{\cdot}10^{-4}$ | 0.97 | 29 |
| tartan (sports) | 13 | $1.0{\cdot}10^{-2}$ | $1.0{\cdot}10^{-4}$ | 0.97 | 30 |
| artifical turf (sports) | 14 | $1.0{\cdot}10^{-2}$ | $1.0{\cdot}10^{-4}$ | 0.94 | 31 |
| clay (sports) | 15 | $1.0{\cdot}10^{-2}$ | $1.0{\cdot}10^{-4}$ | 0.98 | 32 |

**Table A4.** Thermal conductivity $\lambda_{T,i}$ (in W m$^{-1}$ K$^{-1}$) and heat capacity $(\rho C)_i$ (in J K$^{-1}$) of sub-surface layer $i$ according to pavement_type classification. The underlined values are preliminary. The subscript indicates the respective pavement layer. The pavement layers are defined between depth: $0-0.01\,$m, $0.01-0.03\,$m, $0.03-0.07\,$m, $0.07-0.15\,$m, $0.15-0.3\,$m, and $0.3-0.5\,$m.

| Class | $\lambda_{T,1}$ | $\lambda_{T,2}$ | $\lambda_{T,3}$ | $\lambda_{T,4}$ | $\lambda_{T,5}$ | $\lambda_{T,6}$ | $(\rho C)_1$ | $(\rho C)_2$ | $(\rho C)_3$ | $(\rho C)_4$ | $(\rho C)_5$ | $(\rho C)_6$ |
|---|---|---|---|---|---|---|---|---|---|---|---|---|
| 1 | 0.75 | 0.75 | 0.75 | 0.75 | 0.75 | 0.75 | $1.94 \cdot 10^6$ | $1.94 \cdot 10^6$ | $1.94 \cdot 10^6$ | $1.94 \cdot 10^6$ | $1.94 \cdot 10^6$ | $1.94 \cdot 10^6$ |
| 2 | 0.75 | 0.75 | 0.75 | 0.75 | 0.75 | 0.75 | $1.94 \cdot 10^6$ | $1.94 \cdot 10^6$ | $1.94 \cdot 10^6$ | $1.94 \cdot 10^6$ | $1.94 \cdot 10^6$ | $1.94 \cdot 10^6$ |
| 3 | 0.89 | 0.89 | 0.89 | 0.89 | 0.89 | 0.89 | $1.76 \cdot 10^6$ | $1.76 \cdot 10^6$ | $1.76 \cdot 10^6$ | $1.76 \cdot 10^6$ | $1.76 \cdot 10^6$ | $1.76 \cdot 10^6$ |
| 4 | 1.00 | 1.00 | 1.00 | 1.00 | 1.00 | 1.00 | $1.94 \cdot 10^6$ | $1.94 \cdot 10^6$ | $1.94 \cdot 10^6$ | $1.94 \cdot 10^6$ | $1.94 \cdot 10^6$ | $1.94 \cdot 10^6$ |
| 5 | 1.00 | 1.00 | 1.00 | 1.00 | 1.00 | 1.00 | $1.94 \cdot 10^6$ | $1.94 \cdot 10^6$ | $1.94 \cdot 10^6$ | $1.94 \cdot 10^6$ | $1.94 \cdot 10^6$ | $1.94 \cdot 10^6$ |
| 6 | 1.00 | 1.00 | 1.00 | 1.00 | 1.00 | 1.00 | $1.94 \cdot 10^6$ | $1.94 \cdot 10^6$ | $1.94 \cdot 10^6$ | $1.94 \cdot 10^6$ | $1.94 \cdot 10^6$ | $1.94 \cdot 10^6$ |
| 7 | 1.00 | 1.00 | 1.00 | 1.00 | 1.00 | 1.00 | $1.94 \cdot 10^6$ | $1.94 \cdot 10^6$ | $1.94 \cdot 10^6$ | $1.94 \cdot 10^6$ | $1.94 \cdot 10^6$ | $1.94 \cdot 10^6$ |
| 8 | 0.7 | 0.7 | 0.7 | 0.7 | 0.7 | 0.7 | $1.94 \cdot 10^6$ | $1.94 \cdot 10^6$ | $1.94 \cdot 10^6$ | $1.94 \cdot 10^6$ | $1.94 \cdot 10^6$ | $1.94 \cdot 10^6$ |
| 9 | 1.00 | 1.00 | 1.00 | 1.00 | 1.00 | 1.00 | $1.94 \cdot 10^6$ | $1.94 \cdot 10^6$ | $1.94 \cdot 10^6$ | $1.94 \cdot 10^6$ | $1.94 \cdot 10^6$ | $1.94 \cdot 10^6$ |
| 10 | 1.00 | 1.00 | 1.00 | 1.00 | 1.00 | 1.00 | $1.94 \cdot 10^6$ | $1.94 \cdot 10^6$ | $1.94 \cdot 10^6$ | $1.94 \cdot 10^6$ | $1.94 \cdot 10^6$ | $1.94 \cdot 10^6$ |
| 11 | 1.00 | 1.00 | 1.00 | 1.00 | 1.00 | 1.00 | $1.94 \cdot 10^6$ | $1.94 \cdot 10^6$ | $1.94 \cdot 10^6$ | $1.94 \cdot 10^6$ | $1.94 \cdot 10^6$ | $1.94 \cdot 10^6$ |
| 12 | 1.00 | 1.00 | 1.00 | 1.00 | 1.00 | 1.00 | $1.94 \cdot 10^6$ | $1.94 \cdot 10^6$ | $1.94 \cdot 10^6$ | $1.94 \cdot 10^6$ | $1.94 \cdot 10^6$ | $1.94 \cdot 10^6$ |
| 13 | 1.00 | 1.00 | 1.00 | 1.00 | 1.00 | 1.00 | $1.94 \cdot 10^6$ | $1.94 \cdot 10^6$ | $1.94 \cdot 10^6$ | $1.94 \cdot 10^6$ | $1.94 \cdot 10^6$ | $1.94 \cdot 10^6$ |
| 14 | 1.00 | 1.00 | 1.00 | 1.00 | 1.00 | 1.00 | $1.94 \cdot 10^6$ | $1.94 \cdot 10^6$ | $1.94 \cdot 10^6$ | $1.94 \cdot 10^6$ | $1.94 \cdot 10^6$ | $1.94 \cdot 10^6$ |
| 15 | 1.00 | 1.00 | 1.00 | 1.00 | 1.00 | 1.00 | $1.94 \cdot 10^6$ | $1.94 \cdot 10^6$ | $1.94 \cdot 10^6$ | $1.94 \cdot 10^6$ | $1.94 \cdot 10^6$ | $1.94 \cdot 10^6$ |

The layers 1-6 have widths of 0.01, 0.02, 0.04, 0.06, 0.14, and 0.26 m, respectively.

**Table A5.** Water classification parameters according to `water_type`. Underlined values are preliminary. $z_0$ and $z_{0,\mathrm{h}}$ are the roughness lengths for momentum and heat, respectively, and $\epsilon$ is the surface emissivity.

| Description | Class | Water temperature | $z_0$ | $z_{0,\mathrm{h}}$ | $\epsilon$ | albedo_type |
|---|---|---|---|---|---|---|
| | | K | m | m | | |
| lake | 1 | 283 | 0.01 | 0.001 | 0.99 | 1 |
| river | 2 | 283 | 0.01 | 0.001 | 0.99 | 1 |
| ocean | 3 | 283 | 0.01 | 0.001 | 0.99 | 1 |
| pond | 4 | 283 | 0.01 | 0.001 | 0.99 | 1 |
| fountain | 5 | 283 | 0.01 | 0.001 | 0.99 | 1 |

**Table A6.** Soil classification parameters according to `soil_type`. $\alpha_{\mathrm{vG}}$, $l_{\mathrm{vG}}$, $n_{\mathrm{vG}}$ are Van Genuchten parameters, $\gamma_{\mathrm{w,sat}}$ is the hydraulic conductivity at saturation, $m_{\mathrm{sat}}$, $m_{\mathrm{fc}}$, $m_{\mathrm{wilt}}$, and $m_{\mathrm{res}}$ are the volumetric soil moistures at saturation, at field capacity, at wilting point, and the residual soil moisture, respectively.

| Soil texture | Class | $\alpha_{\mathrm{vG}}$ | $l_{\mathrm{vG}}$ | $n_{\mathrm{vG}}$ | $\gamma_{\mathrm{w,sat}}$ | $m_{\mathrm{sat}}$ | $m_{\mathrm{fc}}$ | $m_{\mathrm{wilt}}$ | $m_{\mathrm{res}}$ |
|---|---|---|---|---|---|---|---|---|---|
| | | | | | $\mathrm{m\ s^{-1}}$ | $\mathrm{m^3\ m^{-3}}$ | $\mathrm{m^3\ m^{-3}}$ | $\mathrm{m^3\ m^{-3}}$ | $\mathrm{m^3\ m^{-3}}$ |
| coarse | 1 | 3.83 | 1.150 | 1.38 | $6.94 \cdot 10^{-6}$ | 0.403 | 0.244 | 0.059 | 0.025 |
| medium | 2 | 3.14 | -2.342 | 1.28 | $1.16 \cdot 10^{-6}$ | 0.439 | 0.347 | 0.151 | 0.010 |
| medium-fine | 3 | 0.83 | -0.588 | 1.25 | $0.26 \cdot 10^{-6}$ | 0.430 | 0.383 | 0.133 | 0.010 |
| fine | 4 | 3.67 | -1.977 | 1.10 | $2.87 \cdot 10^{-6}$ | 0.520 | 0.448 | 0.279 | 0.010 |
| very fine | 5 | 2.65 | 2.500 | 1.10 | $1.74 \cdot 10^{-6}$ | 0.614 | 0.541 | 0.335 | 0.010 |
| organic | 6 | 1.30 | 0.400 | 1.20 | $1.20 \cdot 10^{-6}$ | 0.766 | 0.663 | 0.267 | 0.010 |

**Table A7.** Surface albedo classification parameters according to `albedo_type`. Underlined values are preliminary

| Class | Description | broadband | longwave (near-infrared) | shortwave (visible) |
|---|---|---|---|---|
| ocean | 1 | 0.06 | 0.06 | 0.06 |
| mixed farming, tall grassland | 2 | 0.19 | 0.28 | 0.09 |
| tall/medium grassland | 3 | 0.23 | 0.33 | 0.11 |
| evergreen shrubland | 4 | 0.23 | 0.33 | 0.11 |
| short grassland/meadow/shrubland | 5 | 0.25 | 0.34 | 0.14 |
| evergreen needleleaf forest | 6 | 0.14 | 0.22 | 0.06 |
| mixed deciduous forest | 7 | 0.17 | 0.27 | 0.06 |
| deciduous forest | 8 | 0.19 | 0.31 | 0.06 |
| tropical evergreen broadleaved forest | 9 | 0.14 | 0.22 | 0.06 |
| medium/tall grassland/woodland | 10 | 0.18 | 0.28 | 0.06 |
| desert, sandy | 11 | 0.43 | 0.51 | 0.35 |
| desert, rocky | 12 | 0.32 | 0.40 | 0.24 |
| tundra | 13 | 0.19 | 0.27 | 0.10 |
| land ice | 14 | 0.77 | 0.65 | 0.90 |
| sea ice | 15 | 0.77 | 0.65 | 0.90 |
| snow | 16 | 0.82 | 0.70 | 0.95 |
| bare soil | 17 | 0.08 | 0.08 | 0.08 |
| asphalt/concrete mix | 18 | 0.17 | 0.17 | 0.17 |
| asphalt (asphalt concrete) | 19 | 0.17 | 0.17 | 0.17 |
| concrete (Portland concrete) | 20 | 0.30 | 0.30 | 0.30 |
| sett | 21 | 0.17 | 0.17 | 0.17 |
| paving stones | 22 | 0.17 | 0.17 | 0.17 |
| cobblestone | 23 | 0.17 | 0.17 | 0.17 |
| metal | 24 | 0.17 | 0.17 | 0.17 |
| wood | 25 | 0.17 | 0.17 | 0.17 |
| gravel | 26 | 0.17 | 0.17 | 0.17 |
| fine gravel | 27 | 0.17 | 0.17 | 0.17 |
| pebblestone | 28 | 0.17 | 0.17 | 0.17 |
| woodchips | 29 | 0.17 | 0.17 | 0.17 |
| tartan (sports) | 30 | 0.17 | 0.17 | 0.17 |
| artificial turf (sports) | 31 | 0.17 | 0.17 | 0.17 |
| clay (sports) | 32 | 0.17 | 0.17 | 0.17 |
| building | 33 | 0.17 | 0.17 | 0.17 |

**Appendix B: Look-up tables for various input data sources**

**Table B1.** CLC classes to PALM vegetation type.

| CLC class number | description | PALM vegetation type ID |
|---|---|---|
| 111 | Continuous urban fabric | 1 |
| 112 | Discontinuous urban fabric | 1 |
| 121 | Industrial or commercial units | 1 |
| 122 | Road and rail networks and associated land | 1 |
| 123 | Port areas | 1 |
| 124 | Airports | 3 |
| 131 | Mineral extraction sites | 1 |
| 132 | Dump sites | 1 |
| 133 | Construction sites | 1 |
| 141 | Green urban areas | 3 |
| 142 | Sport and leisure facilities | 3 |
| 211 | Non-irrigated arable land | 2 |
| 221 | Vineyards | 2 |
| 222 | Fruit tree and berry plantations | 2 |
| 231 | Pasture, meadows and other perma-nent grasslands under agricultural use | 3 |
| 242 | Complex cultivation patterns | 2 |
| 243 | Land principally occupied by agricul-ture, with significant areas of natural vegetation | 2 |
| 311 | Broad-leaved forest | 7 |
| 312 | Coniferous forest | 4 |
| 313 | Mixed forest | 17 |
| 321 | Natural grassland | 8 |
| 322 | Moors and heathland | 14 |
| 324 | Transitional woodland/shrub | 16 |
| 331 | Beaches, dunes and sand plains | 1 |
| 332 | Bare rock | 1 |
| 333 | Sparsely vegetated areas | 1 |
| 334 | Burnt areas | 1 |
| 335 | Glaciers and perpetual snow | 13 |
| 411 | Inland marshes | 14 |
| 412 | Peatbogs | 14 |
| 421 | Coastal salt marshes | 14 |
| 423 | Intertidal flats | 14 |

**Table B2.** OSM land use classes to PALM vegetation type. "255" is a fill value, assigned to non-vegetation pixels.

| OSM label | PALM vegetation type ID |
|---|---|
| allotments | 2 |
| cemetery | 3 |
| commercial | 255 |
| farm | 2 |
| forest | 17 |
| grass | 3 |
| heath | 3 |
| industrial | 255 |
| meadow | 3 |
| military | 1 |
| orchard | 3 |
| park | 3 |
| quarry | 1 |
| recreation ground | 3 |
| residential | 255 |
| retail | 255 |
| scrub | 16 |
| vineyard | 2 |

**Table B3.** Biotope type groups of the biotope map Hamburg to PALM vegetation type

| Biotope group | Description (German) | approx. translation (English) | PALM vegetation type ID |
|---|---|---|---|
| A | Ruderale und halbruderale Krautflur | (partly) bare soil | 3 |
| B | Biotopkomplexe der Siedlungsflächen | biotopes of build up areas | 1 |
| E | Biotopkomplexe der Freizeit-, Erholungs-, Grünanlagen | biotopes of recreational areas | 3 |
| G | Grünland - (§) (FFH 6510) | lowland hay meadows | 3 |
| H | Gebüsche und Kleingehölze - (§) (FFH 91E0*) | alluvial forests | 16 |
| K | Küstenbiotope - (§) (FFH 1140) | mudflats and sandflats not covered by seawater at low tide | 3 |
| L | Biotope landwirtschaftlich genutzter Flächen | biotopes of agricultural areas | 2 |
| M | Hoch- und Übergangsmoore - (§) (FFH 7110*) | active raised bogs | 14 |
| N | Biotope der Sümpfe und Niedermoore (gehölzfrei) - (§) (FFH 6431) | biotopes of marshes and bogs | 14 |
| O | Offenbodenbiotope | biotopes of bare soil | 1 |
| T | Heiden, Borstgrasrasen, Magerrasen - (§) (FFH 4030) | dry heaths | 3 |
| V | Biotopkomplexe der Verkehrsflächen | biotopes of traffic areas | 3 |
| W | Wald | forest | 17 |
| Y | Biotope vegetationsarmer Flächen im Siedlungsbereich mit Spontanvegetation | biotopes of barely vegetated build up areas with spontanious vegegetation | 1 |
| Z | Vegetationsbestimmte Habitatstrukturen besiedelter Bereiche | vegetation defined habitat structures of build up areas | 1 |

**Table B4.** OSM pavement labels and according PALM pavement types

| OSM pavement label | PALM pavement type | OSM pavement label | PALM pavement type |
|---|---|---|---|
| artificial turf | 14 | metal | 7 |
| asphalt | 2 | metal grid | 7 |
| asphalt:lanes | 2 | mud | 15 |
| asphalt;sett | 2 | paved | 1 |
| clay | 15 | paving stones | 5 |
| cobblestone | 6 | paving stones:30 | 5 |
| cobblestone:flattened | 6 | pebblestone | 11 |
| cobblestone;asphalt | 6 | sand | 10 |
| compacted | 16 | sealed | 1 |
| concrete | 3 | sett | 4 |
| concrete:lanes | 3 | sett;paving stones | 4 |
| concrete:plates | 3 | stone | 6 |
| dirt | 15 | stone:plates | 6 |
| earth | 15 | tartan | 13 |
| fine gravel | 10 | undefined | 1 |
| grass | 1 | unknown pavement | 1 |
| grass paver | 1 | unpaved | 10 |
| gravel | 9 | wood | 8 |
| gravel:tracks | 9 | woodchips | 12 |
| ground | 15 | | |

**Table B5.** OSM road types and corresponding PALM pavement type. The buffer width in this table is only used to convert the line objects into areas if no value is indicated for street width in the OSM data. If the number of lanes is indicated, the street width listed in Tab.B6 is applied.

| OSM road type | buffer width (m) | PALM pavement type |
| --- | --- | --- |
| cycleway | 2.5 | asphalt |
| footway | 2.5 | paving stones |
| living street | 5.5 | paving stones |
| path | 1.5 | undefined |
| primary | 7.5 | asphalt |
| primary link | 7.5 | asphalt |
| residential | 5.5 | asphalt |
| secondary | 6.5 | asphalt |
| secondary link | 6.5 | asphalt |
| service | 5.5 | asphalt |
| tertiary | 5.5 | asphalt |
| tertiary link | 5.5 | asphalt |
| track | 2.6 | undefined |
| trunk | 8 | asphalt |
| trunk link | 8 | asphalt |
| unclassified | 5.5 | asphalt |
| motorway | 0 | asphalt |
| motorway link | 0 | asphalt |

**Table B6.** Assumed road width when in OSM the number of lanes is indicated

| Nr of lanes | buffer width (m) |
| --- | --- |
| 1 | 4 |
| 2 | 9 |
| 3 | 12 |
| 4 | 16 |
| 5 | 20 |
| 6 | 24 |

Table B7: Berlin ISU5 (Informationssystem Stadt und Umwelt, eng.: Informationsystem City and Surroundings) land use descriptions to PALM building type. The building function can be R = residential, O = other, X = no building. The age classes refer to the building period before 1951 (1), between 1951 and 2000 (2) and after 2000 (3). The combination of the function and the building age according to Tab. A2 results in the PALM building type.

| ISU5 land use description | Function | Age | Building type |
|---|---|---|---|
| Allotment garden | R | 2 | 2 |
| Fallow area | X | 0 | 0 |
| Commercial and industrial area, large-scale retail, sparse development | O | 2 | 5 |
| Commercial and industrial area, large-scale retail, dense development | O | 2 | 5 |
| Security and order | O | 2 | 5 |
| Body of water | X | 0 | 0 |
| Heterogeneous inner-city mixed development, post-war gap closure | R | 2 | 2 |
| De-cored block development, post-war gap closure | R | 2 | 2 |
| New school (built after 1945) | O | 2 | 5 |
| Culture | O | 2 | 5 |
| Closed block development, rear courtyard (1870s - 1918), 5-storey | R | 1 | 1 |
| Dense block development, closed rear courtyard (1870s- 1918), 5 - 6-storey | R | 1 | 1 |
| City square / promenade | X | 0 | 0 |
| Block-edge development with large quadrangles (1920s - 1940s), 2 - 5-storey | R | 1 | 1 |
| Park / green space | X | 0 | 0 |
| Railway station and railway ground, without track area | X | 0 | 0 |
| Free row development, landscaped residential greenery (1950s - 1970s), 2 - 6-storey | R | 2 | 2 |
| Old school (built before 1945) | O | 1 | 4 |
| Parking area | X | 0 | 0 |
| Church | O | 1 | 4 |
| Track area | X | 0 | 0 |

| ISU5 land use description | Function | Age | Building type |
|---|---|---|---|
| Children's day care centre | O | 2 | 5 |
| Core area | R | 2 | 2 |
| University and research | O | 2 | 5 |
| Cemetery | X | 0 | 0 |
| Administrative | O | 2 | 5 |
| Non-residential mixed use area, dense development | O | 2 | 5 |
| Utility area | O | 2 | 5 |
| Other and miscellaneous public facility / special use area | O | 2 | 5 |
| Sport facility, uncovered | X | 0 | 0 |
| Rental-flat buildings of the 1990s and later | R | 2 | 2 |
| Non-residential mixed use area, sparse development | O | 2 | 5 |
| Forest | X | 0 | 0 |
| Densification in single-family home area, mixed development with yard and semi-private greening (1870s to present) | R | 1 | 4 |
| Other youth facility | O | 2 | 5 |
| Detached single-family homes with yards | R | 2 | 2 |
| Large estate with tower high-rise buildings (1960s - 1990s), 4 - 11-storey and more | R | 2 | 2 |
| Row houses and duplex with yards | R | 2 | 2 |
| Sport facility, covered | O | 2 | 5 |
| Villas and town villas with park-like gardens (mostly 1870s- 1945) | R | 1 | 1 |
| Other traffic area | X | 0 | 0 |
| Hospital | O | 2 | 5 |
| Tree nursery / horticulture | X | 0 | 0 |
| Parallel row buildings with architectural green strips (1920s - 1930s), 2 - 5 storey | R | 1 | 1 |
| Weekend cottage and allotment-garden-type area | R | 2 | 2 |

| ISU5 land use description | Function | Age | Building type |
|---|---|---|---|
| Closed and semi-open block development, decorative and garden courtyard (1870s - 1918), 4-storey | R | 1 | 1 |
| Construction site | X | 0 | 0 |
| Mixed development, semi-open and open shed courtyard, 2 - 4-storey | O | 2 | 5 |
| Village-like mixed development | O | 2 | 5 |
| Agriculture | X | 0 | 0 |
| Airport | O | 2 | 5 |
| Camping ground | X | 0 | 0 |

Table B8: ALKIS land use descriptions to PALM building type. The building function can be R = residential, O = other, X = no building. The combination of the function and the building age according to Tab. A2 results in the PALM building type.

| Summarized description | ALKIS ID | land use description (German) | Function |
|---|---|---|---|
| Residential buildings | 1000 | Wohngebäude | R |
| | 1010 | Wohnhaus | R |
| | 1020 | Wohnheim | R |
| | 1022 | Seniorenheim | R |
| | 1024 | Studenten-, Schülerwohnheim | R |
| | 1025 | Schullandheim | R |
| Mixed use buildings (incl. residential) | 1100 | Gemischt genutztes Gebäude mit Wohnen | R |
| | 1110 | Wohngebäude mit Gemeinbedarf | R |
| | 1120 | Wohngebäude mit Handel und Dienstleistungen | R |
| | 1121 | Wohn- und Verwaltungsgebäude | R |
| | 1122 | Wohn- und Bürogebäude | R |
| | 1123 | Wohn- und Geschäftsgebäude | R |
| | 1130 | Wohngebäude mit Gewerbe und Industrie | R |
| | 1131 | Wohn- und Betriebsgebäude | R |
| Mixed used agricultural buildings | 1210 | Land- und forstwirtschaftliches Wohngebäude | R |
| | 1220 | Land- und forstwirtschaftliches Wohn- und Betriebsgebäude | R |
| | 1222 | Wohn- und Wirtschaftsgebäude | R |

| Summarized description | ALKIS ID | land use description (German) | Function |
|---|---|---|---|
| | 1223 | Forsthaus | O |
| Recreational buildings | 1310 | Gebäude zur Freizeitgestaltung | O |
| | 1311 | Ferienhaus | R |
| | 1312 | Wochenendhaus | R |
| | 1313 | Gartenhaus | R |
| Commercial buildings (incl. shops, malls) | 2000 | Gebäude für Wirtschaft oder Gewerbe | O |
| | 2010 | Gebäude für Handel und Dienstleistungen | O |
| | 2020 | Bürogebäude | O |
| | 2030 | Kreditinstitut | O |
| | 2040 | Versicherung | O |
| | 2050 | Geschäftsgebäude | O |
| | 2051 | Kaufhaus | O |
| | 2052 | Einkaufszentrum | O |
| | 2053 | Markthalle | O |
| | 2054 | Laden | O |
| | 2055 | Kiosk | O |
| | 2060 | Messehalle | O |
| Hotel, Restaurant etc. | 2071 | Hotel, Motel, Pension | O |
| | 2072 | Jugendherberge | O |
| | 2074 | Campingplatzgebäude | O |
| | 2080 | Gebäude für Bewirtung | O |
| | 2081 | Gaststätte, Restaurant | O |
| | 2083 | Kantine | O |
| Cinema, Casino, ... | 2090 | Freizeit- und Vergnügungsstätte | O |
| | 2092 | Kino | O |
| | 2094 | Spielkasino | O |
| Commercial and industrial buildings, incl. factories, warehouses, garages | 2100 | Gebäude für Gewerbe und Industrie | O |
| | 2110 | Produktionsgebäude | O |
| | 2111 | Fabrik | O |
| | 2112 | Betriebsgebäude | O |
| | 2120 | Werkstatt | O |
| | 2130 | Tankstelle | O |
| | 2131 | Waschstraße, Waschanlage, Waschhalle | O |

| Summarized description | ALKIS ID | land use description (German) | Function |
|---|---|---|---|
| | 2140 | Gebäude für Vorratshaltung | O |
| | 2141 | Kühlhaus | O |
| | 2142 | Speichergebäude | O |
| | 2143 | Lagerhalle, Lagerschuppen, Lagerhaus | O |
| | 2150 | Speditionsgebäude | O |
| | 2160 | Gebäude für Forschungszwecke | O |
| | 2180 | Gebäude für betriebliche Sozialeinrichtung | O |
| | 2200 | Sonstiges Gebäude für Gewerbe und Industrie | O |
| | 2213 | Schöpfwerk | O |
| | 2310 | Gebäude für Handel und Dienstleistung mit Wohnen | O |
| | 2320 | Gebäude für Gewerbe und Industrie mit Wohnen | O |
| | 2400 | Betriebsgebäude zu Verkehrsanlagen (allgemein) | O |
| | 2410 | Betriebsgebäude für Straßenverkehr | O |
| | 2411 | Straßenmeisterei | O |
| | 2412 | Wartehalle | O |
| | 2420 | Betriebsgebäude für Schienenverkehr | O |
| | 2422 | Lokschuppen, Wagenhalle | O |
| | 2430 | Betriebsgebäude für Flugverkehr | O |
| | 2431 | Flugzeughalle | O |
| | 2440 | Betriebsgebäude für Schiffsverkehr | O |
| | 2441 | Werft (Halle) | O |
| | 2443 | Betriebsgebäude zur Schleuse | O |
| | 2444 | Bootshaus | O |
| | 2460 | Gebäude zum Parken | O |
| | 2461 | Parkhaus | O |
| | 2462 | Parkdeck | O |
| | 2463 | Garage | O |
| | 2464 | Fahrzeughalle | O |
| | 2465 | Tiefgarage | X |
| Utility buildings | 2500 | Gebäude zur Versorgung | O |
| Energy utility buildings | 2501 | Gebäude zur Energieversorgung | O |

| Summarized description | ALKIS ID | land use description (German) | Function |
|---|---|---|---|
| Water utility buildings | 2510 | Gebäude zur Wasserversorgung | O |
| | 2512 | Pumpstation | O |
| | 2513 | Wasserbehälter | X |
| | 2520 | Gebäude zur Elektrizitätsversorgung | O |
| | 2521 | Elektrizitätswerk | O |
| | 2522 | Umspannwerk | O |
| | 2523 | Umformer | X |
| | 2528 | Turbinenhaus | O |
| | 2529 | Kesselhaus | O |
| Heating, gas and waste water utility buildings | 2540 | Gebäude für Fernmeldewesen | O |
| | 2560 | Gebäude an unterirdischen Leitungen | O |
| | 2570 | Gebäude zur Gasversorgung | O |
| | 2580 | Heizwerk | O |
| | 2590 | Gebäude zur Versorgungsanlage | O |
| | 2600 | Gebäude zur Entsorgung | O |
| | 2611 | Gebäude der Kläranlage | O |
| | 2612 | Toilette | O |
| Waste buildings | 2620 | Gebäude zur Abfallbehandlung | O |
| | 2621 | Müllbunker | O |
| | 2622 | Gebäude zur Müllverbrennung | O |
| Buildings for agriculture and forestry | 2700 | Gebäude für Land- und Forstwirtschaft | O |
| | 2720 | Land- und forstwirtschaftliches Betriebsgebäude | O |
| | 2721 | Scheune | O |
| | 2723 | Schuppen | O |
| | 2724 | Stall | O |
| | 2726 | Scheune und Stall | O |
| | 2728 | Reithalle | O |
| | 2729 | Wirtschaftsgebäude | O |
| | 2740 | Treibhaus, Gewächshaus | O |
| | 2741 | Treibhaus | O |
| | 2742 | Gewächshaus, verschiebbar | O |

| Summarized description | ALKIS ID | land use description (German) | Function |
|---|---|---|---|
| Public buildings, schools and cultural buildings | 3000 | Gebäude für öffentliche Zwecke | O |
| | 3010 | Verwaltungsgebäude | O |
| | 3012 | Rathaus | O |
| | 3013 | Post | O |
| | 3014 | Zollamt | O |
| | 3015 | Gericht | O |
| | 3016 | Botschaft, Konsulat | O |
| | 3019 | Finanzamt | O |
| | 3020 | Gebäude für Bildung und Forschung | O |
| | 3021 | Allgemein bildende Schule | O |
| | 3022 | Berufsbildende Schule | O |
| | 3023 | Hochschulgebäude (Fachhochschule, Universität) | O |
| | 3024 | Forschungsinstitut | O |
| | 3030 | Gebäude für kulturelle Zwecke | O |
| | 3031 | Schloss | O |
| | 3032 | Theater, Oper | O |
| | 3033 | Konzertgebäude | O |
| | 3034 | Museum | O |
| | 3035 | Rundfunk, Fernsehen | O |
| | 3036 | Veranstaltungsgebäude | O |
| | 3037 | Bibliothek, Bücherei | O |
| Religious buildings | 3040 | Gebäude für religiöse Zwecke | O |
| | 3041 | Kirche | O |
| | 3042 | Synagoge | O |
| | 3043 | Kapelle | O |
| | 3044 | Gemeindehaus | O |
| | 3045 | Gotteshaus | O |
| | 3046 | Moschee | O |
| | 3048 | Kloster | O |
| Hospitals and simlilar buildings | 3050 | Gebäude für Gesundheitswesen | O |
| | 3051 | Krankenhaus | O |
| | 3052 | Heilanstalt, Pflegeanstalt, Pflegestation | O |
| | 3053 | Ärztehaus, Poliklinik | O |

| Summarized description | ALKIS ID | land use description (German) | Function |
|---|---|---|---|
| Buildings for social use, day care, also buildings of emergency services and cementries | 3060 | Gebäude für soziale Zwecke | O |
| | 3061 | Jugendfreizeitheim | O |
| | 3062 | Freizeit-, Vereinsheim, Dorfgemeinschafts-, Bürgerhaus | O |
| | 3065 | Kinderkrippe, Kindergarten, Kindertagesstätte | O |
| | 3070 | Gebäude für Sicherheit und Ordnung | O |
| | 3071 | Polizei | O |
| | 3072 | Feuerwehr | O |
| | 3073 | Kaserne | O |
| | 3074 | Schutzbunker | O |
| | 3075 | Justizvollzugsanstalt | O |
| | 3080 | Friedhofsgebäude | O |
| | 3081 | Trauerhalle | O |
| | 3082 | Krematorium | O |
| | 3090 | Empfangsgebäude | O |
| Trafic related buildings (e.g. train station, airport) | 3091 | Bahnhofsgebäude | O |
| | 3092 | Flughafengebäude | O |
| | 3097 | Gebäude zum Busbahnhof | O |
| | 3098 | Empfangsgebäude Schifffahrt | O |
| Public buildings with residential use | 3100 | Gebäude für öffentliche Zwecke mit Wohnen | O |
| Recreational buildings | 3200 | Gebäude für Erholungszwecke | O |
| Sport related buildings (Stadiums, Swimming pools etc. ) | 3210 | Gebäude für Sportzwecke | O |
| | 3211 | Sport-, Turnhalle | O |
| | 3212 | Gebäude zum Sportplatz | O |
| | 3220 | Badegebäude | O |
| | 3221 | Hallenbad | O |
| | 3222 | Gebäude im Freibad | O |
| | 3230 | Gebäude im Stadion | O |
| | 3240 | Gebäude für Kurbetrieb | O |
| Touristic buildings | 3260 | Gebäude im Zoo | O |
| | 3270 | Gebäude im botanischen Garten | O |
| | 3281 | Schutzhütte | O |
| | 3290 | Touristisches Informationszentrum | O |

| Summarized description | ALKIS ID | land use description (German) | Function |
|---|---|---|---|
| Unknown | 9998 | unbekannt | U |
| Club homes | 11161 | Jugendhaus | O |
| | 11162 | Waldheim | R |
| | 11491 | Vereinsheim | O |
| Elevator | 12359 | Aufzug | X |
| Retention basin | 12619 | Rückhaltebecken | X |
| Towers and chimneys | 12629 | Kamin | X |
| | 19702 | Wasserturm | X |
| | 19703 | Kirchturm, Glockenturm | O |
| | 19704 | Aussichtsturm | O |
| | 19705 | Sende-, Funk-, Fernmeldeturm | X |
| | 19706 | Stadt-, Torturm | X |
| Canopy | 19901 | Überdachung | X |
| Basement | 19902 | Unterkellerung | X |

**Table B9.** OSM values to PALM water type. "255" is a fill value, assigned to non-water pixels.

| OSM label | PALM water type ID |
| --- | --- |
| canal | 2 |
| ditch | 2 |
| drain | 255 |
| fountain | 5 |
| lake | 1 |
| pond | 4 |
| reflection-pool | 4 |
| reservoir | 1 |
| river | 2 |
| riverbank | 2 |
| stream | 2 |
| (null) | 255 |
| Brack | 4 |
| Teich | 4 |
| See | 1 |
| Weiher | 4 |
| Moor | 255 |

**Table B10.** CLC classes to PALM water type

| CLC class number | description | PALM water type ID |
|---|---|---|
| 511 | Water courses | 2 |
| 512 | Water bodies | 1 |
| 521 | Coastal lagoons | 3 |
| 522 | Estuaries | 3 |
| 523 | Sea and ocean | 3 |

Table B11: Biotope types of Hamburg to PALM water type

| Summarized description | Biotope abbreviation | Description (German) | PALM water type ID |
|---|---|---|---|
| Streams | FBA | Bach, ausgebaut | 2 |
| | FBM | Bach, naturnah mit Beeinträch-tigungen/Verbauungen - (§) (FFH 3260) | 2 |
| | FBR | Bach, weitgehend naturnah | 2 |
| | FBS | Aufgestauter Bachabschnitt | 2 |
| | FBT | Bach-Altarm | 2 |
| Rivers | FFA | Fluss, ausgebaut | 2 |
| | FFF | Flachwasserbereiche der Elbe | 2 |
| | FFM | Fluss, naturnah mit Beeinträch-tigungen/Verbauungen | 2 |
| | FFR | Fluss, weitgehend naturnah | 2 |
| | FFS | Aufgestauter Flussabschnitt | 2 |
| | FFT | Fluss-Altarm | 2 |
| Ditch | FG | Graben mit Stillgewässercharakter | 4 |
| | FGA | Nährstoffarmer Graben mit Stillgewässercharakter - | 4 |
| | FGM | Graben mittlerer Nährstoffgehalte mit Stillgewässercharakter - (§) | 4 |
| | FGR | Nährstoffreicher Graben mit Stillgewässer-charakter | 4 |
| | FGV | Nährstoffreicher Graben mit Stillgewässer-charakter | 4 |
| | FGX | Abwassergraben | 2 |
| Inner harbour | FH | Hafenbecken | 1 |
| Channel | FK | Kanal | 2 |
| | FLH | Wettern, Hauptgraben | 2 |
| | FLM | Graben mittlerer Nährstoffgehalte mit Fließgewässer-charakter | 2 |
| | FLR | Nährstoffreicher Graben mit Fließgewässer-charakter | 2 |
| Source areas | FQ | Quellbereich - | 4 |
| | FQG | Tümpelquelle - § | 4 |
| | FQS | Tümpelquelle - § | 4 |
| River strand, river bank | FSO | Flussstrand, gestört - (§) | 255 |
| | FSV | Flussstrand, naturnah - | 255 |
| | FSW | Strandwall am Elbufer - | 255 |

| Summarized description | Biotope abbreviation | Description (German) | PALM water type ID |
|---|---|---|---|
| (Mud)flats | FWB | Flusswatt mit Pioniervegetation - | 2 |
| | FWO | Flusswatt, ohne Bewuchs | 2 |
| | FWP | Priel | 2 |
| | FWV | Tideröhricht | 2 |
| | FWX | Verbautes Elbufer mit naturnahen Vegetations-elementen | 255 |
| | FWZ | Sonstige naturnahe Flächen im Wasserwechsel-bereich der tide beein-flussten Flussunterläufe | 255 |
| Small water basins, eutrophic and near-natural | SEA | Abbaugewässer, klein, naturnah, nährstoffreich | 4 |
| | SEB | Brack, naturnah, nährstoffreich | 4 |
| | SED | Bombentrichter, naturnah, nährstoffreich | 4 |
| | SEF | Altwasser, klein, naturnah | 4 |
| | SEG | Angelegte Kleingewässer, klein, naturnah, nährstoffreich - § FFH 3150 18 | 4 |
| | SEN | Natürliches, nährstoffreiches Kleingewässer | 4 |
| | SEO | Nährstoffreiche Kleingewässer ohne Bewuchs | 4 |
| | SER | Naturnahes, nährstoffreiches Regenrückhalte-becken | 4 |
| | SES | Nährstoffreiche Kleingewässer mit artenarmem Bewuchs | 4 |
| | SET | Teich, nährstoffreich, naturnah | 4 |
| | SEW | Weidekuhle, nährstoffreich, naturnah | 4 |
| | SEY | Beregnungsbecken mit naturnahen Elementen | 4 |
| | SEZ | Sonstiges, naturnahes, nährstoffreiches Kleingewässer | 4 |
| Artificial lakes | SGA | Abbaugewässer, Baggersee, groß | 1 |
| | SGF | Altwasser, groß | 1 |
| | SGT | Staugewässer, groß | 1 |
| | SGZ | Sonstiges Stillgewässer, groß | 1 |
| Small water bodies, near-natural and nutrient-poor | SOA | Abbaugewässer, naturnah, nährstoffarm | 4 |
| | SOG | Angelegtes Kleingewässer, naturnah, nährstoffarm | 4 |
| | SOM | Moorgewässer, naturnah, nährstoffarm | 4 |
| | SON | Kleingewässer natürlicher Entstehung, naturnah, nährstoffarm | 4 |
| | SOT | Teich, nährstoffarm, naturnah | 4 |
| Ponds | STA | Ackertümpel | 4 |
| | STG | Wiesen- oder Weidetümpel | 4 |

| Summarized description | Biotope abbreviation | Description (German) | PALM water type ID |
|---|---|---|---|
| | STR | Rohbodentümpel | 4 |
| | STW | Waldtümpel | 4 |
| | STZ | Sonstiger Tümpel | 4 |
| | SX | Naturfernes Stillgewässer | 4 |
| | SXA | Naturfernes Abbaugewässer | 4 |
| | SXB | Sonstiges Brack | 4 |
| | SXG | Naturfernes Ziergewässer | 4 |
| | SXK | Klärteich, Absetzbecken | 4 |
| Artificial water bodies | SXL | Löschwasserbecken, naturfern | 4 |
| | SXP | Fischteich, naturfern | 4 |
| | SXR | Rückhaltebecken, naturfern | 4 |
| | SXT | Teich, naturfern | 4 |
| | SXY | Beregnungsbecken, naturfern | 4 |
| | SXZ | Sonstiges, naturfernes Wasserbecken | 4 |

*Author contributions.* BM, MS and FKS designed the input data requirements for PALM. MS implemented the data input and the PALM-internal data processing. BM developed the `palm_csd` code and created the static driver in the supplements based on the input data of Berlin provided by WH and JZ. WH and JZ generated the input data for Berlin, Stuttgart and Hamburg shown in the manuscript. WH and BM prepared the manuscript with contributions of MS, JZ, DP, CB and TE

5 *Competing interests.* The authors declare that they have no conflict of interest.

*Acknowledgements.* The work presented in this paper was funded by the German Federal Ministry of Education and Research (BMBF) under grant 01LP1601 within the framework of Research for Sustainable Development (FONA; www.fona.de), which is greatly acknowledged. We thank R. Kapp (Amt für Umweltschutz) and J. Oberdorfer (Stadtvermessungsamt) of the Landeshauptstadt Stuttgart, Germany for providing the municipal data of Stuttgart. We would like to thank the two anonymous reviewers for their helpful comments on the manuscript.

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
