# Peer review of "Geospatial input data for the PALM model system 6.0: model requirements, data sources, and processing"

_Geoscientific Model Development, 2019_

## Referee Comment (RC1) · Anonymous Referee #1 · 18 May 2020

General Summary and Comments This manuscript describes a semi-automated system for preparing a set of environmental input data for the PALM model system. The PALM model is designed to simulate micro- and mesoscale flow dynamics in urban environments. Examples of the system for deriving parameter sets for three German cities are provided. The input parameters (building, soil, water, road, and vegetation properties) appear to be thoroughly defined, extensive, and complete. Overall, the manuscript is well-written, clearly presented, and recognizes previous work in this area. Although, please see the Technical Comments below. The main aspects of the manuscript that require improvement are related to the need to provide a more complete description of the PALM model to provide context for the rest of the paper, to

include some examples of input parameters in individual sections, and the lack of any discussion regarding whether the model has been run/evaluated/validated with these parameter sets (see Specific Comments below for more details).

Specific Comments

1. Section 1, Lines 7-15: I think this paragraph should be expanded to provide more information on what PALM is and how it is structured and operates (for instance, what spatial resolution does the model typically operate at?), including some background on how it has been applied in past studies, with references. In addition, what capabilities does the model have (i.e., later in the paper there is reference to "embedded chemistry model" and "embedded multi-agent system for urban residents" and "nested simulation setup" and "compute the energy demand of single buildings". This information only slowly becomes apparent as one reads through the paper. Maybe there is an existing graphic from a previous paper that could help the reader visualize the environment that the model is attempting to represent (e.g., what types of surfaces are represented?). This would be valuable to provide some context for the rest of the paper describing the input data. This may be particularly important since the Maronga et al. 2019 paper that apparently describes PALM is listed as submitted? 2. Section 2: It would be useful and clarifying to provide one or two concrete examples for each of the subsections describing the input requirements, so that the reader doesn't necessarily have to go to the tables to understand each section (particularly since these tables are in the appendix). For instance, what are some of the water body types and parameters described in section 2.6.3, what are some of the soil types and parameters described in section 2.7, etc. 3. I didn't see any indication that the model has actually been run with this set of data for the three example cities? Was it run and the output verified/validated? What were the problems encountered and what were the overall results?

Technical Comments

1. Abstract: Is PALM an acronym? I don't see it defined in the abstract. 2. Abstract,

[Figure]

Line 5: Suggest rewording or removing "standardized, hereafter called PALM input data standard". Maybe "...follows a standardized format, hereafter called the PALM input data standard". On the other hand it may not be necessary to call this out in the abstract. Later in the paper this is referred to as "the so-called PALM input data standard". 3. Page 2, Line 4: Suggest changing "inevitable" to "necessary". 4. Page 2, Line 18: Suggest changing "fitting to the" to "fit the". 5. Page 2, Line 30: Suggest changing "is therefore also conform with" with "and therefore also conforms to the" 6. Page 3, Line 7: There is a reference here to Table 12 (not Table 1). Is this typical table numbering for GMD, i.e., to consecutively number tables starting with the ones in the Appendix and then continue with tables referred to in the main paper? 7. Page 3, Line 10: Suggest changing "extent" to "extend". 8. Page 4, Line 12: Are root fractions vertically resolved? 9. Page 4, Line 12: Only much later in the paper is it apparent that BAD is not used in the current implementation. It would be useful to note that here. 10. Page 4, Line 24: Do you mean zt = 0? 11. Page 5, Line 30: I don't understand "in case the energy balance for building surface should be solved". Shouldn't it always be solved for? 12. Page 6, Line 3: Are these building types appropriate only for a specific region (e.g., country), or are they generally applicable globally? 13. Page 6, Line 4: Suggest changing "build" to "built". 14. Page 6, Line 17: In general, if vegetation is not present, is bare soil parameterized? 15. Page 6, Line 18: Not resolved by the vertical numerical grid? 16. Page 7, Line 10: What types of water surface and parameters are available? Is there a table to be referenced here? 17. Page 10, Line 6: Are the pre-processing scripts referred to here publically available? Or are these simply the same thing as the palm_csd? Are these scripts general enough such that they could be applied to other cites, at least in Germany? 18. Page 11, Line 15: "7, 8"? 19. Page 13: Is there any additional data that is available, now or in the future, that could alleviate this problem (no information on restoration and heat insulation actions)? 20. Page 14: What threshold of NDVI was used? Was it proven to be accurate? Please spell out this first use of the term NDVI (normalized difference vegetation index) (as noted in the next section). 21. Page 15: Please define here what you mean by intensive and extensive

green roofs. Only later on does it become apparent that extensive means trees growing over the roofs. 22. Page 17, Line 4: Units for LAI? m2 leaf area per m2 of ground area? Does LAI then vary with time in the model? If so, at what temporal resolution? What part of the model handles this? 23. Page 17, Line 9: What is an "image granule"? 24. Page 17, Line 16: Please spell out IDL. Interactive Data Language? Reference? 25. Page 19, Line 1: It seem like pavement thickness would be required to accurately model heat transfer. Is this considered/available? 26. Page 20: Line 1: How are street type and street crossings used in the model? What is the difference between pavement type and street type? 27. Page 21, Line 3: Change "respcective" to "respective". 28. Page 21, Line 14: Change "in into" to "into". 29. Page 21, Line 27: Change "his" to "their". 30. Tables A1, A3, A5, A6 captions: Please describe each of the symbols given here. In Table A1, what is IFS? In Table A3, what are the units for these parameters? 31. Table A4: Are there values for different depths in the soil? What are the depths? 32. Table A7: By "longwave" and "shortwave" do you mean near-infrared and visible, respectively? 33. Table B2: Is "255" a missing or fill value or something else? 34. Table B3: Is an English translation useful or appropriate here? 35. Table B5: Is buffer width the same as street width? I don't see any cases where "the number of lanes is indicated". Are these in meters? 36. Table B6: Are these in meters? 37. Table B8: Is an English translation useful or appropriate here? 38. Table B9: Is "255 a missing or fill value or something else? 39. Table B11: What is "Biotope"? Change "abbriviation" to "abbreviation". Is an English translation useful or appropriate here? 40. Table 13: This table refers to tables in the supplements, but the only supplement file appears to be the netcdf file that represents the input for the static driver. For example, what is "Tab.S10 in the Supplements"? Please provide more information for readers.

---

## Referee Comment (RC2) · Anonymous Referee #2 · 21 Jun 2020

General comments

This paper presents how to describe the local environment for the PALM model system 6.0. The different modules, standards and possible data-sources are in this sense clearly and exhaustively presented. Three German cities are used to illustrate the different steps in the data collection and that allows to appreciate a certain spatial and intra-parameter variability. An important set of good quality figures and tables describing the input parameters accompany the text.

Although I am sure of the interest of such an article for PALM users, I think it would gain interest for the whole community of modelers if the work is discussed more in depth

regarding the state of the art. Even if authors recognizes previous work in this area no analysis or discussion is made regarding it. In its current state the article is closer to a technical report than to a scientific paper. My main comment would be regarding this point.

Specific comments

The content of the paper is well written but there are numerous section crossings and this should be avoided to minimize repetition. I agree with reviewer 1, even if this paper is published in a special issue accompanied by an article on the PALM model it must be self-sufficient. Some contextual information about the model and the eventual numerical simulations where these data were used, are missed in the current version. Some subsections are very concise (for example 2.6.3 for water bodies), please complete them or combine them with others sections. The paragraph in Pg3. L30, is diffcult to understand. An extra figure or workflow could be useful to guide the reader. As a general rule, please integrate the figures after their citation.

Technical corrections Pg2. L1-2, "as well" is used twice Pg7. L7, there is a "-" after the word vegetation. Pg9. L8, please update the sentence "Nevertheless, a German-wide coverage is planned for the end of 2019" Pg10. Sec4. Here you can perhaps cite and discuss your work regarding the paper Bocher, E.; Petit, G.; Bernard, J. & Palominos, S. A geoprocessing framework to compute urban indicators: The MApUCE tools chain Urban Climate, 2018, 24, 153 - 174 Pg10. Sec 4.1, it is not clear the horizontal resolution of the airborne lidar. Is it 1m? Pg 12. L20, a dot is missed after "pavement)". Pg14. L3, "As an area wide approach satellite..." This sentence it's not clear to me, perhaps a grammar revision is needed. Pg14. L9, I think a coma is needed after "For the vegetation type layer municipal". Pg15. L28, a ")" is missed.

Fig1. It's difficult to compare the three terrains as the legend is not homogeneus. Fig3. Please include buildings types directly in the legend. Fig5. It seems that the method identifies vegetation over the building path. Please explain and discuss the limits.

---

## Author Comment (AC1) · 31 Aug 2020

First of all, we want to thank the two anonymous reviewers for their valuable comments.

Summarizing, the reviewers hat the following main recommendations:

1) Include a more complete description of the PALM model including the use of the model to make the paper easier to read independent of the PALM overview paper by Maronga et al. (2020).

2) Increase the scientific value of the manuscript by discussing the content of the manuscript towards other approaches in literature, e.g. regarding input data and model

evaluation/validation.

3) The paragraph on the different Levels of Detail (LOD) of PALM is difficult to understand.

In order to address these we changed the following to the manuscript:

1) We added a section in the introduction giving more background information on the PALM model.

2) To increase the scientific value of the paper, we included a section (Sect. 6) showing a PALM simulation using the static driver created for this paper. The simulation result is followed by a discussion of the experiences on the input data by related studies which focus on evaluating PALM (partly with the data presented in this paper). We included also a discussion towards other data frameworks for urban climate analysis (LZC, WUDAPT, MApUCE) in section 4.

3) To illustrate the concept of LOD in PALM, we added an illustration using LAI as an example for a city quarter in Berlin, Germany and show the difference between LOD0, LOD1 and LOD2 for this vegetation_par.

In addition to these general recommendations, which resulted in the largest changes of the manuscript, the reviewers also provided numerous specific and technical comments. They resulted in smaller changes in the manuscript, but helped to improve the article a lot, for which we want to cordially thank the reviewers. Detailed answers to these comments are provided in the attached pdf.

Please also note the supplement to this comment:
https://gmd.copernicus.org/preprints/gmd-2019-355/gmd-2019-355-AC1-supplement.pdf

**Supplement:**

GMD 2019 355

**Geospatial input data for the PALM model system 6.0: modelrequirements, data sources, and processing**
Wieke Heldens, Cornelia Burmeister, Farah Kanani-Sühring, Björn Maronga, Dirk Pavlik,Matthias Sühring, Julian Zeidler, and Thomas Esch

**Final response by the authors**

First of all, we want to thank the two anonymous reviewers for their valuable comments. Summarizing, the reviewers hat the following main recommendations:

1) Include a more complete description of the PALM model including the use of the model to make the paper easier to read independent of the PALM overview paper by Maronga et al. (2020).
2) Increase the scientific value of the manuscript by discussing the content of the manuscript towards other approaches in literature, e.g. regarding input data and model evaluation/validation.
3) The paragraph on the different Levels of Detail (LOD) of PALM is difficult to understand.

In order to address these we changed the following to the manuscript:

1) We added a section in the introduction giving more background information on the PALM model.
2) To increase the scientific value of the paper, we included a section (Sect. 6) showing a PALM simulation using the static driver created for this paper. The simulation result is followed by a discussion of the experiences on the input data by related studies which focus on evaluating PALM (partly with the data presented in this paper). We included also a discussion towards other data frameworks for urban climate analysis (LZC, WUDAPT, MApUCE) in section 4.
3) To illustrate the concept of LOD in PALM, we added an illustration using LAI as an example for a city quarter in Berlin, Germany and show the difference between LOD0, LOD1 and LOD2 for this vegetation_par.

In addition to these general recommendations, which resulted in the largest changes of the manuscript, the reviewers also provided numerous specific and technical comments. They resulted in smaller changes in the manuscript, but helped to improve the article a lot, for which we want to cordially thank the reviewers. Detailed answers to these comments are provided below.

**Anonymous Referee #1 – Author Response**

**General Summary and Comments**

This manuscript describes a semi-automated system for preparing a set of environmental input data for the PALM model system. The PALM model is designed to simulate micro- and mesoscale flow dynamics in urban environments. Examples of the system for deriving parameter sets for three German cities are provided. The input parameters (building, soil, water, road, and vegetation properties) appear to be thoroughly defined, extensive, and complete. Overall, the manuscript is well-written, clearly presented, and recognizes previous work in this area. Although, please see the Technical Comments below. The main aspects of the manuscript that require improvement are related to the need to provide a more complete description of the PALM model to provide context for the rest of the paper, to include some examples of input parameters in individual sections, and the lack of any discussion regarding whether the model has been run/evaluated/validated with these parameter sets (see Specific Comments below for more details).

**A.** Thank you for these valuable comments. We answered them in detail after the according specific comments below.

**Specific Comments**

**1.** Section 1, Lines 7-15: I think this paragraph should be expanded to provide more information on what PALM is and how it is structured and operates (for instance, what spatial resolution does the model typically operate at?), including some background on how it has been applied in past studies, with references. In addition, what capabilities does the model have (i.e., later in the paper there is reference to "embedded chemistry model" and "embedded multi-agent system for urban residents" and "nested simulation setup" and "compute the energy demand of single buildings". This information only slowly becomes apparent as one reads through the paper. Maybe there is an existing graphic from a previous paper that could help the reader visualize the environment that the model is attempting to represent (e.g., what types of surfaces are represented?). This would be valuable to provide some context for the rest of the paper describing the input data. This may be particularly important since the Maronga et al. 2019 paper that apparently describes PALM is listed as submitted?

**A1.** We included an additional section in the introduction, so that it is more clear which capabilities the model has and for what kind of studies it can be used. Also, since March 2020, the paper of Maronga et al. describing PALM6.0 is officially published. The citation is updated accordingly.

**2.** Section 2: It would be useful and clarifying to provide one or two concrete examples for each of the subsections describing the input requirements, so that the reader doesn't necessarily have to go to the tables to understand each section (particularly since these tables are in the appendix). For instance, what are some of the water body types and parameters described in section 2.6.3, what are some of the soil types and parameters described in section 2.7, etc.

**A2:** We added examples of the classes/types for clarification and a reference to the according table in the Appendix for faster identification.

**3.** I didn't see any indication that the model has actually been run with this set of data for the three example cities? Was it run and the output verified/validated? What were the problems encountered and what were the overall results?

**A3.** We added an additional section (section 6) where we show the static driver (which is provided in the supplement) that was generated with the data presented in the paper and show simulation results generated with this data. Additionally, we discussed experiences of other studies with this input data and discuss what the most important aspects are to generate sound simulation results with PALM.

**Technical Comments**

**1.** Abstract: Is PALM an acronym? I don't see it defined in the abstract.

**A1.** PALM used to be an abbreviation for Parallelized Large-eddy Simulation Model but is now an independent name (see Maronga et al., 2020).

**2.** Abstract, Line 5: Suggest rewording or removing "standardized, hereafter called PALM input data standard". Maybe "...follows a standardized format, hereafter called the PALM input data standard". On the other hand it may not be necessary to call this out in the abstract. Later in the paper this is referred to as "the so-called PALM input data standard".

**A2.** We changed the sentence to "...and follows a standardized format."

**3.** Page 2, Line 4: Suggest changing "inevitable" to "necessary".

**A3.** Wording changed as suggested.

**4.** Page 2, Line 18: Suggest changing "fitting to the" to "fit the".

**A4.** Wording changed as suggested.

**5.** Page 2, Line 30: Suggest changing "is therefore also conform with" with "and therefore also conforms to the"

**A5.** Wording changed as suggested.

**6.** Page 3, Line 7: There is a reference here to Table 12 (not Table 1). Is this typical table numbering for GMD, i.e., to consecutively number tables starting with the ones in the Appendix and then continue with tables referred to in the main paper?

**A6.** The numbering by accident went wrong in Latex as the tables were placed at the end of the manuscript. The tables are now included directly in the text and start with #1.

**7.** Page 3, Line 10: Suggest changing "extent" to "extend".

**A7.** Wording changed as suggested.

**8.** Page 4, Line 12: Are root fractions vertically resolved?

**A8.** The root fractions are distributed over the individual vertical soil layers and must sum up to 1. We added a respective note.

**9.** Page 4, Line 12: Only much later in the paper is it apparent that BAD is not used in the current implementation. It would be useful to note that here.

**A9.** We added a note.

**10.** Page 4, Line 24: Do you mean zt = 0?

**A10.** Yes, this should be zt. We corrected it accordingly.

**11.** Page 5, Line 30: I don't understand "in case the energy balance for building surface should be solved". Shouldn't it always be solved for?

**A11.** Yes. The sentence was imprecise. We rephrased it to: "To solve the energy balance at building surfaces (Resler et al., 2017; Maronga et al., 2020), information on the type of thebuildings must be provided. To solve the energy balance at building surfaces appropriately, various wall material and surface properties must be known, e.g….."

**12.** Page 6, Line 3: Are these building types appropriate only for a specific region (e.g., country), or are they generally applicable globally?

**A12.** The provided wall and surface parameters are assumed to be valid for Germany, but still have to be checked and likely changed for other building styles. We added a not accordingly. As from other studies (e.g. Belda et al. 2020) the importance of albedo and thermal conductivity of building

walls on the simulation results is emphasized as well, we included this topic also in the discussion in Section 6 and the conclusion.

**13.** Page 6, Line 4: Suggest changing "build" to "built".

**A13.** Wording changed as suggested.

**14.** Page 6, Line 17: In general, if vegetation is not present, is bare soil parameterized?

**A14.** No, a vegetation type (including bare soil), water type, building (incl. bridges) or a pavement type has to be assigned. In our processing chain, we therefore checked for such pixels and filled them according to a ranking of available data (see section 4.2), which mostly resulted in either vegetation or pavement. If after these filling iterations there is still a gap, a default value is inserted, which currently is bare soil indeed. We added a comment in Section 4.2 to clarify this.

**15.** Page 6, Line 18: Not resolved by the vertical numerical grid?

**A15.** With this it is meant that the vegetation height is smaller than the vertical grid spacing. We added a note.

**16.** Page 7, Line 10: What types of water surface and parameters are available? Is there a table to be referenced here?

**A16.** Table A5 in the appendix contains this information. We included the cross reference to this table and extended also the text with information on the available water types and parameters.

**17.** Page 10, Line 6: Are the pre-processing scripts referred to here publically available? Or are these simply the same thing as the palm csd? Are these scripts general enough such that they could be applied to other cites, at least in Germany?

**A17:** The scripts and programs used to process the input data are currently not publicly available as they use some commercial and or internal functionality. However we are currently in the process of maturing them and plan to make them publicly available with the PALM Model in the future. We are currently in the process of (trying to) generalize them so they can also be applied to other cities. However this strongly depends on the stability of input data sources. So e.g. the scripts based on OSM are easily transferable (the user might have to add mappings for pavement types unknown in Germany), but especially building 3D data is very heterogeneous. We added a not in the manuscript to inform all readers.

**18.** Page 11, Line 15: "7, 8"?

**A18.** A separating comma was inserted between the footnotes.

**19.** Page 13: Is there any additional data that is available, now or in the future, that could alleviate this problem (no information on restoration and heat insulation actions)?

**A19.** Currently there is no additional data available. However, there are some ongoing initiatives which might help in the future. We changed the text in Section 4.3.2. to give some information on these initiatives: "In Germany there is no cadastral information on restoration and heat insulation actions for individual buildings, so that the age of building construction used for building classification is often a rather poor proxy for the thermodynamic properties of buildings. Theoretically, energetic information on buildings exists in Germany, described in the energy performance certificate of each residential building. However, this information is not generally accessible. Currently, a research activity funded by the German Federal Ministry of Economic Affairs and Energy is developing a data base on energetic characteristics of non-residential buildings (ENOB DataNWG (https://datanwg.de/home/aktuelles/ ). Thus, for small areas where high quality simulations are desired, field surveys or drone surveys still are required. For the large area applications in this study however, this was not possible. Instead, we aimed at identifying the classes defined in Tab. B2. "

**20.** Page 14: What threshold of NDVI was used? Was it proven to be accurate? Please spell out this first use of the term NDVI (normalized difference vegetation index) (as noted in the next section).

**A20.** The term NDVI is explained in Section 4.4.1. now (and section 4.4.2 is changed accordingly). The thresholds used are explained in the text now. They are based on the extensive and intensive vegetation classes, where extensive vegetation is assumed to be short grass (class 3) and intensive vegetation is assumed to be deciduous broadleaf (class 7). For the small areas where this data set is used for (to fill gaps in the pavement layer) the accuracy has proven to be sufficient.

**21.** Page 15: Please define here what you mean by intensive and extensive green roofs. Only later on does it become apparent that extensive means trees growing over the roofs.

**A21.** Green roofs are categorized into intensive (with trees and park-like vegetation) and extensive (most common type, with succulents, mosses etc.) green roofs. We added an explanatory sentence.

**22.** Page 17, Line 4: Units for LAI? m2 leaf area per m2 of ground area? Does LAI then vary with time in the model? If so, at what temporal resolution? What part of the model handles this?

**A22.** Yes, the unit of LAI is m2 leaf area per m2 of ground area and therefore LAI is dimensionless. LAI does not vary with time in PALM, but it is meaning-full to use the LAI that represents the vegetation state at the simulated day. For that reason, we prepared different LAI data sets to select from. We added 2 sentences to explain this in Section 4.4.5.: "LAI is defined as m2 leaf area per m2 of ground area and is therefore dimensionless. The LAI varies largely over the phenological cycle. Therefore it is important to approximate the state of the vegetation of the day to be simulated in PALM as best as possible in the input data."

**23.** Page 17, Line 9: What is an "image granule"?

**A23.** Sentinel 2 images are divided into smaller, manageable pieces which are called "image granules" or tiles ("For Level-1C and Level-2A, the granules, also called tiles, are 100x100 km2 ortho-images in UTM/WGS84 projection." https://eos.com/sentinel-2/) The sentence was rephrased to "Depending on the study area up to three Sentinel image tiles (so called image granules)..." to make this understandable for people outside the field of remote sensing.

**24.** Page 17, Line 16: Please spell out IDL. Interactive Data Language? Reference?

**A24.** Yes, it stands for Interactive Data language. We spelled it out and referred to the software company.

**25.** Page 19, Line 1: It seem like pavement thickness would be required to accurately model heat transfer. Is this considered/available?

**A25.** In fact, the model assumes by default that the pavement covers the 6 uppermost soil layers. We added a note in Section 2.7 ("Soil classification"): "By default the soil layers have widths of (from top to bottom) of 0.01, 0.02, 0.04, 0.06, 0.14, 0.26, 0.54, and 1.86 m, where pavement layers are by default assumed to cover the six uppermost layers (see Tab. A4). Note that the vertical composition of materials used in road construction varies a lot and is usually unknown. The current implementation thus should be considered to be a first guess."

**26.** Page 20: Line 1: How are street type and street crossings used in the model? What is the difference between pavement type and street type?

**A26.** At the moment, input data for street_type and street_crossing is used by PALM's embedded chemistry model (Maronga et al. 2020, Khan et al. submitted) to parametrize emissions of chemical compounds during the diurnal cycle. This information was also added to the manuscript. The pavement type describes the material of the surface (e.g. asphalt), whereas the street type describes the use of the road (e.g. high way) and thus gives an indication on the amount of emission traffic on the street produces.

**27.** Page 21, Line 3: Change "respcective" to "respective".

**A27.** Corrected.

**28.** Page 21, Line 14: Change "in into" to "into".

**A28.** Corrected.

**29.** Page 21, Line 27: Change "his" to "their".

**A29.** Corrected.

**30.** Tables A1, A3, A5, A6 captions: Please describe each of the symbols given here. In Table A1, what is IFS? In Table A3, what are the units for these parameters?

**A30.** Missing information was added to the table captions.

**31.** Table A4: Are there values for different depths in the soil? What are the depths?

**A31.** The index number indicates the respective layer within the soil. We added the soil layer widths to the caption. See also comment above.

**32.** Table A7: By "longwave" and "shortwave" do you mean near-infrared and visible, respectively?

**A32.** Yes. We added a note to the caption.

**33.** Table B2: Is "255" a missing or fill value or something else?

**A33.** "255" is indeed a fill value, as these OSM classes should not be assigned to any vegetation class. We included a note in the caption.

**34.** Table B3: Is an English translation useful or appropriate here?

**A34.** Although the classes are very specific and likely only used in Germany only, we attempted a translation for the interested user.

**35.** Table B5: Is buffer width the same as street width? I don't see any cases where "the number of lanes is indicated". Are these in meters?

**A35.** No, this is not the same. "Street width" and "number of lanes" are both attributes that can be set in OSM (but are not always available for all objects). However, the caption is imprecise. We corrected it to: "The buffer width in this table is only used to convert the line objects into areas if no value is indicated for street width in the OSM data. If the number of lanes is indicated, the street width listed in Tab.B6 is applied." The buffer width is in meters. We added this to the table.

**36.** Table B6: Are these in meters?

**A36.** Yes. We added this to the table.

**37.** Table B8: Is an English translation useful or appropriate here?

**A37.** As this classification is always only available in German for German areas, a detailed translation seems not especially relevant. However, for interested readers, we grouped the classes and made summarized translations.

**38.** Table B9: Is "255 a missing or fill value or something else?

**A38.** "255" is indeed a fill value. We added a note.

**39.** Table B11: What is "Biotope"? Change "abbriviation" to "abbreviation". Is an English translation useful or appropriate here?

**A39**. A biotope type is comparable to a habitat type. Both words exist in English (according to leo.org.) We choose the word biotope because it is close to the German word "Biotop", which is the name of this kind of map in Germany (Biotope Maps = Biotopkartierung). Similar to Table B8, a detailed translation is relevant, as the classes are very detailed. However, we added here a summarized translation as well.

**40**. Table A3: This table refers to tables in the supplements, but the only supplement file appears to be the netcdf file that represents the input for the static driver. For example, what is "Tab.S10 in the Supplements"? Please provide more information for readers.

**A40.** This was simply a mistake, meant was Table A4.

**Anonymous Referee #2 – Author response**

**General comments**

**1.** This paper presents how to describe the local environment for the PALM model system 6.0. The different modules, standards and possible data-sources are in this sense clearly and exhaustively presented. Three German cities are used to illustrate the different steps in the data collection and that allows to appreciate a certain spatial and intra-parameter variability. An important set of good quality figures and tables describing the input parameters accompany the text. Although I am sure of the interest of such an article for PALM users, I think it would gain interest for the whole community of modelers if the work is discussed more in depth regarding the state of the art. Even if authors recognizes previous work in this area no analysis or discussion is made regarding it. In its current state the article is closer to a technical report than to a scientific paper. My main comment would be regarding this point.

**A1:** Thank you for these valuable comments. To increase the scientific value of the paper, we included a section (Sect. 6) showing and reflecting a simulation using the static driver created for this paper. We included also a discussion towards other data frameworks for urban climate analysis (LZC, WUDAPT, MApUCE) in section 4.

**Specific comments**

**2.** The content of the paper is well written but there are numerous section crossings and this should be avoided to minimize repetition.

**A2.** We went through the paper and tried to remove repetitions where possible. However, the structure of the input data cannot avoid repetitions all together. We briefly considered another organization of the paper, in which for each parameter in section 2 follows directly the input data processing (currently section 3) of this parameter. However, many decisions on data selection and processing are the result of the way the static driver is created as a whole (including on how parameters are treated in combination). Therefore we decided to describe first all requirements in detail and then, with this knowledge in the back of the readers head, describe the input data processing.

**3.** I agree with reviewer 1, even if this paper is published in a special issue accompanied by an article on the PALM model it must be self-sufficient. Some contextual information about the model and the eventual numerical simulations where these data were used, are missed in the current version.

**A3:** We added a section to the introduction describing PALM in more detail, so the paper is easier to read on its own.

**4.** Some subsections are very concise (for example 2.6.3 for water bodies), please complete them or combine them with others sections.

**A4.** We completed the short sections. However, they still remain less detailed then others, mainly because the parameters (such as water bodies) are straight forward to map and data is readily available.

**5.** The paragraph in Pg3. L30, is difficult to understand. An extra figure or workflow could be useful to guide the reader.

**A5.** We added an illustration using LAI for a city quarter in Berlin, Germany. Here the LOD1 LAI field is determined from the vegetation_type classification and then overwritten by LOD2 data via vegetation_pars: "Figure 1 shows this hierarchy exemplary for the LAI based on available data for Berlin, Germany."

[Figure]

**Figure 1.** Illustration of LOD1 and LOD2 hierarchy for the LAI around Berlin Ernst-Reuter-Platz. Panel (a) shows the determined vegetation_type classification, which involves automatic setting of an LAI for all pixels classified as vegetation (LOD1), shown in panel (b). Panel (c) shows LOD2 information of the LAI distribution transferred by the field vegetation_pars.

**6.** As a general rule, please integrate the figures after their citation.

**A6.** We made sure that in our Latex file, the figures are inserted directly in after the passage where they are first mentioned. LaTeX then places the figures on the pages.

**Technical corrections**

**7.** Pg2. L1-2, "as well" is used twice

**A7.** We removed the first instance.

**8.** Pg7. L7, there is a "-" after the word vegetation.

**A8.** We removed the "-"

**9.** Pg9. L8, please update the sentence "Nevertheless, a Germanwide coverage is planned for the end of 2019"

**A9.** The section was updated to "Since 2019, LOD2 building data exists for all German states (Arbeitsgemeinschaft der Vermessungsverwaltungen der Länder der Bundesrepublik Deutschland,2019a, b). However, accessibility and cost vary for each state."

**10.** Pg10. Sec4. Here you can perhaps cite and discuss your work regarding the paper Bocher, E.; Petit, G.; Bernard, J. & Palominos, S. A geoprocessing framework to compute urban indicators: The MApUCE tools chain Urban Climate, 2018, 24, 153 - 174

**A10:** We included a discussion of LCZ, WUDAPT and MApUCE at the beginning of section 4.

**11.** Pg10. Sec 4.1, it is not clear the horizontal resolution of the airborne lidar. Is it 1m?

**A11.** Yes, the horizontal resolution of the DTM created based on the airborne LiDAR data is 1 m. We clarified the text.

**12.** Pg 12. L20, a dot is missed after "pavement)".

**A12.** Added.

**13.** Pg 14. L3, "As an area wide approach satellite…" This sentence it's not clear to me, perhaps a grammar revision is needed.

**A13.** We rephrased the sentence to: "As an area wide approach satellite or airborne imagery provides accurate information on the location of the vegetation. Such remote sensing data can also provide estimates for some vegetation characteristics, but not all required by PALM."

**14.** Pg14. L9, I think a coma is needed after "For the vegetation type layer municipal".

**A14.** Comma added.

**15.** Pg15. L28, a ")" is missed.

**A15.** Added.

**16.** Fig1. It's difficult to compare the three terrains as the legend is not homogeneous.

**A16.** True. However, the legends were chosen to show maximum topography within each city. With identical legends, the range will be too large as Hamburg has terrain heights around 0, and Stuttgart around 600 and no relief within the cities will be visible. Therefore we choose to use different legends for each city and we prefer to keep them that way.

**17.** Fig3. Please include buildings types directly in the legend.

**A17.** We added the building types into the legend.

**18.** Fig5. It seems that the method identifies vegetation over the building path. Please explain and discuss the limits.

**A18**. Yes, the vegetation layers available from the municipalities are often quite coarse and do not always have holes for buildings/paths. Therefore as stated in part (4.2 Surface classification) there is a ranking of "spatial reliability" between the core layers, so that e.g. a building pixel is preferred over a vegetation pixel.

**References**

Arbeitsgemeinschaft der Vermessungsverwaltungen der Länder der Bundesrepublik Deutschland 3D-Gebäudemodelle LoD1: Produktblatt, 2019

Arbeitsgemeinschaft der Vermessungsverwaltungen der Länder der Bundesrepublik Deutschland 3D-Gebäudemodelle LoD2: Produktblatt 2019

Khan, B.; Banzhaf, S.; Chan, E. C.; Forkel, R.; Kanani-Sühring, F.; Ketelsen, K.; Kurppa, M.; Maronga, B.; Mauder, M.; Raasch, S.; Russo, E.; Schaap, M. & Sühring, M. Development of an Atmospheric Chemistry Model coupled to the PALM Model System 6.0: Implementation and First Applications *Geoscientific Model Development Discussions,* submitted.

Maronga, B.; Banzhaf, S.; Burmeister, C.; Esch, T.; Forkel, R.; Fröhlich, D.; Fuka, V.; Gehrke, K. F.; Geletič, J.; Giersch, S.; Gronemeier, T.; Groß, G.; Heldens, W.; Hellsten, A.; Hoffmann, F.; Inagaki, A.; Kadasch, E.; Kanani-Sühring, F.; Ketelsen, K.; Khan, B. A.; Knigge, C.; Knoop, H.; Krč, P.; Kurppa, M.; Maamari, H.; Matzarakis, A.; Mauder, M.; Pallasch, M.; Pavlik, D.; Pfafferott, J.; Resler, J.; Rissmann, S.; Russo, E.; Salim, M.; Schrempf, M.; Schwenkel, J.; Seckmeyer, G.; Schubert, S.; Sühring, M.; von Tils, R.; Vollmer, L.; Ward, S.; Witha, B.; Wurps, H.; Zeidler, J. & Raasch, S. Overview of the PALM model system 6.0 *Geoscientific Model Development,* 2020, 13, 1335-1372

Resler, J.; Krč, P.; Belda, M.; Juruš, P.; Benešová, N.; Lopata, J.; Vlček, O.; Damašková, D.; Eben, K.; Derbek, P.; Maronga, B. & Kanani-Sühring, F. PALM-USM v1.0: A new urban surface model integrated into the PALM large-eddy simulation model *Geoscientific Model Development,* 2017, 10, 3635-3659